# Fourier Features Let Networks Learn High Frequency Functions in Low Dimensional Domains

Matthew Tancik[1]*    Pratul P. Srinivasan[1,2]*    Ben Mildenhall[1]*    Sara Fridovich-Keil[1]

Nithin Raghavan[1]    Utkarsh Singhal[1]    Ravi Ramamoorthi[3]    Jonathan T. Barron[2]    Ren Ng[1]

[1]University of California, Berkeley    [2]Google Research    [3]University of California, San Diego

## Abstract

We show that passing input points through a simple Fourier feature mapping enables a multilayer perceptron (MLP) to learn high-frequency functions in low-dimensional problem domains. These results shed light on recent advances in computer vision and graphics that achieve state-of-the-art results by using MLPs to represent complex 3D objects and scenes. Using tools from the neural tangent kernel (NTK) literature, we show that a standard MLP has impractically slow convergence to high frequency signal components. To overcome this spectral bias, we use a Fourier feature mapping to transform the effective NTK into a stationary kernel with a tunable bandwidth. We suggest an approach for selecting problem-specific Fourier features that greatly improves the performance of MLPs for low-dimensional regression tasks relevant to the computer vision and graphics communities.

## 1 Introduction

A recent line of research in computer vision and graphics replaces traditional discrete representations of objects, scene geometry, and appearance (*e.g.* meshes and voxel grids) with continuous functions parameterized by deep fully-connected networks (also called multilayer perceptrons or MLPs). These MLPs, which we will call "coordinate-based" MLPs, take low-dimensional coordinates as inputs (typically points in $\mathbb{R}^3$) and are trained to output a representation of shape, density, and/or color at each input location (see Figure 1). This strategy is compelling since coordinate-based MLPs are amenable to gradient-based optimization and machine learning, and can be orders of magnitude more compact than grid-sampled representations. Coordinate-based MLPs have been used to represent images [31, 42] (referred to as "compositional pattern producing networks"), volume density [30], occupancy [27], and signed distance [35], and have achieved state-of-the-art results across a variety of tasks such as shape representation [9, 11, 13, 14, 19, 29, 35], texture synthesis [17, 34], shape inference from images [24, 25], and novel view synthesis [30, 32, 38, 41].

We leverage recent progress in modeling the behavior of deep networks using kernel regression with a neural tangent kernel (NTK) [18] to theoretically and experimentally show that standard MLPs are poorly suited for these low-dimensional coordinate-based vision and graphics tasks. In particular, MLPs have difficulty learning high frequency functions, a phenomenon referred to in the literature as "spectral bias" [3, 36]. NTK theory suggests that this is because standard coordinate-based MLPs correspond to kernels with a rapid frequency falloff, which effectively prevents them from being able to represent the high-frequency content present in natural images and scenes.

---

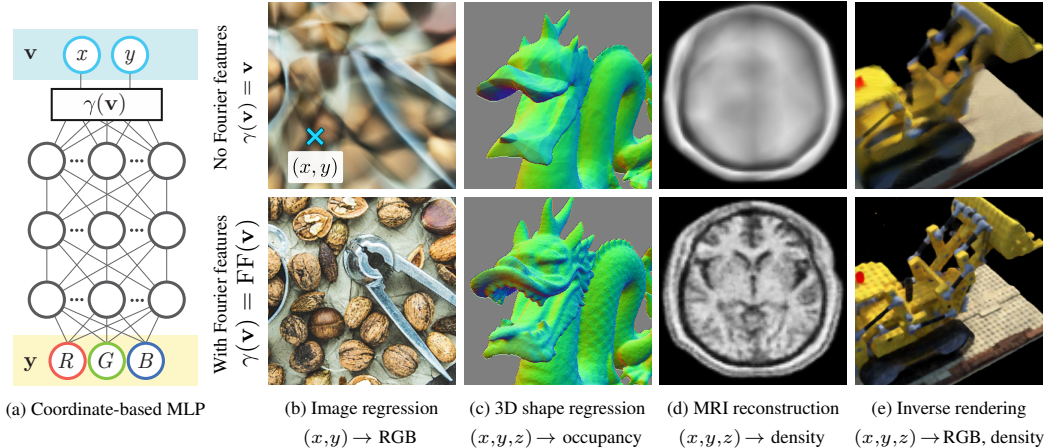

(a) Coordinate-based MLP

(b) Image regression
$(x,y) \rightarrow$ RGB

(c) 3D shape regression
$(x,y,z) \rightarrow$ occupancy

(d) MRI reconstruction
$(x,y,z) \rightarrow$ density

(e) Inverse rendering
$(x,y,z) \rightarrow$ RGB, density

Figure 1: Fourier features improve the results of coordinate-based MLPs for a variety of high-frequency low-dimensional regression tasks, both with direct (b, c) and indirect (d, e) supervision. We visualize an example MLP (a) for an image regression task (b), where the input to the network is a pixel coordinate and the output is that pixel's color. Passing coordinates directly into the network (top) produces blurry images, whereas preprocessing the input with a Fourier feature mapping (bottom) enables the MLP to represent higher frequency details.

A few recent works [30, 48] have experimentally found that a heuristic sinusoidal mapping of input coordinates (called a "positional encoding") allows MLPs to represent higher frequency content. We observe that this is a special case of Fourier features [37]: mapping input coordinates $\mathbf{v}$ to $\gamma(\mathbf{v}) = \left[ a_1 \cos(2\pi \mathbf{b}_1^T \mathbf{v}), a_1 \sin(2\pi \mathbf{b}_1^T \mathbf{v}), \ldots, a_m \cos(2\pi \mathbf{b}_m^T \mathbf{v}), a_m \sin(2\pi \mathbf{b}_m^T \mathbf{v}) \right]^T$ before passing them into an MLP. We show that this mapping transforms the NTK into a stationary (shift-invariant) kernel and enables tuning the NTK's spectrum by modifying the frequency vectors $\mathbf{b}_j$, thereby controlling the range of frequencies that can be learned by the corresponding MLP. We show that the simple strategy of setting $a_j = 1$ and randomly sampling $\mathbf{b}_j$ from an isotropic distribution achieves good performance, and that the scale (standard deviation) of this distribution matters much more than its specific shape. We train MLPs with this Fourier feature input mapping across a range of tasks relevant to the computer vision and graphics communities. As highlighted in Figure 1, our proposed mapping dramatically improves the performance of coordinate-based MLPs. In summary, we make the following contributions:

- We leverage NTK theory and simple experiments to show that a Fourier feature mapping can be used to overcome the spectral bias of coordinate-based MLPs towards low frequencies by allowing them to learn much higher frequencies (Section 4).

- We demonstrate that a random Fourier feature mapping with an appropriately chosen scale can dramatically improve the performance of coordinate-based MLPs across many low-dimensional tasks in computer vision and graphics (Section 5).

## 2 Related Work

Our work is motivated by the widespread use of coordinate-based MLPs to represent a variety of visual signals, including images [42] and 3D scenes [27, 30, 35]. In particular, our analysis is intended to clarify experimental results demonstrating that an input mapping of coordinates (which they called a "positional encoding") using sinusoids with logarithmically-spaced axis-aligned frequencies improves the performance of coordinate-based MLPs on the tasks of novel view synthesis from 2D images [30] and protein structure modeling from cryo-electron microscopy [48]. We analyze this technique to show that it corresponds to a modification of the MLP's NTK, and we show that other non-axis-aligned frequency distributions can outperform this positional encoding.

Prior works in natural language processing and time series analysis [20, 43, 46] have used a similar positional encoding to represent time or 1D position. In particular, Xu *et al.* [46] use random Fourier

features (RFF) [37] to approximate stationary kernels with a sinusoidal input mapping and propose techniques to tune the mapping parameters. Our work extends this by directly explaining such mappings as a modification of the resulting network's NTK. Additionally, we address the embedding of multidimensional coordinates, which is necessary for vision and graphics tasks.

To analyze the effects of applying a Fourier feature mapping to input coordinates before passing them through an MLP, we rely on recent theoretical work that models neural networks in the limits of infinite width and infinitesimal learning rate as kernel regression using the NTK [2, 5, 12, 18, 22]. In particular, we use the analyses from Lee *et al.* [22] and Arora *et al.* [2], which show that the outputs of a network throughout gradient descent remain close to those of a linear dynamical system whose convergence rate is governed by the eigenvalues of the NTK matrix [2, 3, 5, 22, 47]. Analysis of the NTK's eigendecomposition shows that its eigenvalue spectrum decays rapidly as a function of frequency, which explains the widely-observed "spectral bias" of deep networks towards learning low-frequency functions [3, 4, 36].

We leverage this analysis to consider the implications of adding a Fourier feature mapping before the network, and we show that this mapping has a significant effect on the NTK's eigenvalue spectrum and on the corresponding network's convergence properties in practice.

## 3 Background and Notation

To lay the foundation for our theoretical analysis, we first review classic kernel regression and its connection to recent results that analyze the training dynamics and generalization behavior of deep fully-connected networks. In later sections, we use these tools to analyze the effects of training coordinate-based MLPs with Fourier feature mappings.

**Kernel regression.** Kernel regression is a classic nonlinear regression algorithm [44]. Given a training dataset $(\mathbf{X}, \mathbf{y}) = \{(\mathbf{x}_i, y_i)\}_{i=1}^n$, where $\mathbf{x}_i$ are input points and $y_i = f(\mathbf{x}_i)$ are the corresponding scalar output labels, kernel regression constructs an estimate $\hat{f}$ of the underlying function at any point $\mathbf{x}$ as:

$$\hat{f}(\mathbf{x}) = \sum_{i=1}^{n} \left( \mathbf{K}^{-1} \mathbf{y} \right)_i k(\mathbf{x}_i, \mathbf{x}), \tag{1}$$

where $\mathbf{K}$ is an $n \times n$ kernel (Gram) matrix with entries $\mathbf{K}_{ij} = k(\mathbf{x}_i, \mathbf{x}_j)$ and $k$ is a symmetric positive semidefinite (PSD) kernel function which represents the "similarity" between two input vectors. Intuitively, the kernel regression estimate at any point $\mathbf{x}$ can be thought of as a weighted sum of training labels $y_i$ using the similarity between the corresponding $\mathbf{x}_i$ and $\mathbf{x}$.

**Approximating deep networks with kernel regression.** Let $f$ be a fully-connected deep network with weights $\theta$ initialized from a Gaussian distribution $\mathcal{N}$. Theory proposed by Jacot *et al.* [18] and extended by others [2, 3, 22] shows that when the width of the layers in $f$ tends to infinity and the learning rate for SGD tends to zero, the function $f(\mathbf{x}; \theta)$ converges over the course of training to the kernel regression solution using the *neural tangent kernel* (NTK), defined as:

$$k_{\text{NTK}}(\mathbf{x}_i, \mathbf{x}_j) = \mathbb{E}_{\theta \sim \mathcal{N}} \left\langle \frac{\partial f(\mathbf{x}_i; \theta)}{\partial \theta}, \frac{\partial f(\mathbf{x}_j; \theta)}{\partial \theta} \right\rangle. \tag{2}$$

When the inputs are restricted to a hypersphere, the NTK for an MLP can be written as a dot product kernel (a kernel in the form $h_{\text{NTK}}(\mathbf{x}_i^\mathsf{T} \mathbf{x}_j)$ for a scalar function $h_{\text{NTK}} : \mathbb{R} \to \mathbb{R}$).

Prior work [2, 3, 18, 22] shows that an NTK linear system model can be used to approximate the dynamics of a deep network during training. We consider a network trained with an L2 loss and a learning rate $\eta$, where the network's weights are initialized such that the output of the network at initialization is close to zero. Under asymptotic conditions stated in Lee *et al.* [22], the network's output for any data $\mathbf{X}_{\text{test}}$ after $t$ training iterations can be approximated as:

$$\hat{\mathbf{y}}^{(t)} \approx \mathbf{K}_{\text{test}} \mathbf{K}^{-1} \left( \mathbf{I} - e^{-\eta \mathbf{K} t} \right) \mathbf{y}, \tag{3}$$

where $\hat{\mathbf{y}}^{(t)} = f(\mathbf{X}_{\text{test}}; \theta)$ are the network's predictions on input points $\mathbf{X}_{\text{test}}$ at training iteration $t$, $\mathbf{K}$ is the NTK matrix between all pairs of training points in $\mathbf{X}$, and $\mathbf{K}_{\text{test}}$ is the NTK matrix between all points in $\mathbf{X}_{\text{test}}$ and all points in the training dataset $\mathbf{X}$.

**Spectral bias when training neural networks.** Let us consider the training error $\hat{\mathbf{y}}_{\text{train}}^{(t)} - \mathbf{y}$, where $\hat{\mathbf{y}}_{\text{train}}^{(t)}$ are the network's predictions on the training dataset at iteration $t$. Since the NTK matrix $\mathbf{K}$ must be PSD, we can take its eigendecomposition $\mathbf{K} = \mathbf{Q}\boldsymbol{\Lambda}\mathbf{Q}^{\text{T}}$, where $\mathbf{Q}$ is orthogonal and $\boldsymbol{\Lambda}$ is a diagonal matrix whose entries are the eigenvalues $\lambda_i \geq 0$ of $\mathbf{K}$. Then, since $e^{-\eta\mathbf{K}t} = \mathbf{Q}e^{-\eta\boldsymbol{\Lambda}t}\mathbf{Q}^{\text{T}}$:

$$\mathbf{Q}^{\text{T}}(\hat{\mathbf{y}}_{\text{train}}^{(t)} - \mathbf{y}) \approx \mathbf{Q}^{\text{T}}\left(\left(\mathbf{I} - e^{-\eta\mathbf{K}t}\right)\mathbf{y} - \mathbf{y}\right) = -e^{-\eta\boldsymbol{\Lambda}t}\mathbf{Q}^{\text{T}}\mathbf{y}\,. \tag{4}$$

This means that if we consider training convergence in the eigenbasis of the NTK, the $i^{\text{th}}$ component of the absolute error $|\mathbf{Q}^{\text{T}}(\hat{\mathbf{y}}_{\text{train}}^{(t)} - \mathbf{y})|_i$ will decay approximately exponentially at the rate $\eta\lambda_i$. In other words, components of the target function that correspond to kernel eigenvectors with larger eigenvalues will be learned faster. For a conventional MLP, the eigenvalues of the NTK decay rapidly [4, 5, 16]. This results in extremely slow convergence to the high frequency components of the target function, to the point where standard MLPs are effectively unable to learn these components, as visualized in Figure 1. Next, we describe a technique to address this slow convergence by using a Fourier feature mapping of input coordinates before passing them to the MLP.

## 4    Fourier Features for a Tunable Stationary Neural Tangent Kernel

Machine learning analysis typically addresses the case in which inputs are high dimensional points (*e.g.* the pixels of an image reshaped into a vector) and training examples are sparsely distributed. In contrast, in this work we consider *low-dimensional regression* tasks, wherein inputs are assumed to be dense coordinates in a subset of $\mathbb{R}^d$ for small values of $d$ (*e.g.* pixel coordinates). This setting has two significant implications when viewing deep networks through the lens of kernel regression:

1. We would like the composed NTK to be shift-invariant over the input domain, since the training points are distributed with uniform density. In problems where the inputs are normalized to the surface of a hypersphere (common in machine learning), a dot product kernel (such as the regular NTK) corresponds to spherical convolution. However, inputs in our setting are dense in Euclidean space. A Fourier feature mapping of input coordinates makes the composed NTK stationary (shift-invariant), acting as a convolution kernel over the input domain (see Appendix C for additional discussion on stationary kernels).

2. We would like to control the bandwidth of the NTK to improve training speed and generalization. As we see from Eqn. 4, a "wider" kernel with a slower spectral falloff achieves faster training convergence for high frequency components. However, we know from signal processing that reconstructing a signal using a kernel whose spectrum is *too* wide causes high frequency aliasing artifacts. We show in Section 5 that a Fourier feature input mapping can be tuned to lie between these "underfitting' and "overfitting" extremes, enabling both fast convergence and low test error.

**Fourier features and the composed neural tangent kernel.** Fourier feature mappings have been used in many applications since their introduction in the seminal work of Rahimi and Recht [37], which used random Fourier features to approximate an arbitrary stationary kernel function by applying Bochner's theorem. Extending this technique, we use a Fourier feature mapping $\gamma$ to featurize input coordinates before passing them through a coordinate-based MLP, and investigate the theoretical and practical effect this has on convergence speed and generalization. The function $\gamma$ maps input points $\mathbf{v} \in [0, 1)^d$ to the surface of a higher dimensional hypersphere with a set of sinusoids:

$$\gamma(\mathbf{v}) = \left[a_1 \cos(2\pi\mathbf{b}_1^{\text{T}}\mathbf{v}), a_1 \sin(2\pi\mathbf{b}_1^{\text{T}}\mathbf{v}), \ldots, a_m \cos(2\pi\mathbf{b}_m^{\text{T}}\mathbf{v}), a_m \sin(2\pi\mathbf{b}_m^{\text{T}}\mathbf{v})\right]^{\text{T}}\,. \tag{5}$$

Because $\cos(\alpha - \beta) = \cos\alpha\cos\beta + \sin\alpha\sin\beta$, the kernel function induced by this mapping is:

$$k_\gamma(\mathbf{v}_1, \mathbf{v}_2) = \gamma(\mathbf{v}_1)^{\text{T}}\gamma(\mathbf{v}_2) = \sum_{j=1}^{m} a_j^2 \cos\left(2\pi\mathbf{b}_j^{\text{T}}\left(\mathbf{v}_1 - \mathbf{v}_2\right)\right) = h_\gamma(\mathbf{v}_1 - \mathbf{v}_2)\,, \tag{6}$$

$$\text{where } h_\gamma(\mathbf{v}_\Delta) \triangleq \sum_{j=1}^{m} a_j^2 \cos(2\pi\mathbf{b}_j^{\text{T}}\mathbf{v}_\Delta)\,. \tag{7}$$

Note that this kernel is stationary (a function of only the difference between points). We can think of the mapping as a Fourier approximation of a kernel function: $\mathbf{b}_j$ are the Fourier basis frequencies used to approximate the kernel, and $a_j^2$ are the corresponding Fourier series coefficients.

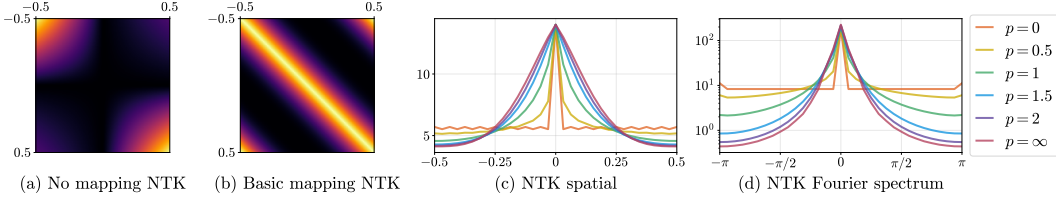

(a) No mapping NTK     (b) Basic mapping NTK     (c) NTK spatial     (d) NTK Fourier spectrum

Figure 2: Adding a Fourier feature mapping can improve the poor conditioning of a coordinate-based MLP's neural tangent kernel (NTK). (a) We visualize the NTK function $k_{\mathrm{NTK}}(x_i, x_j)$ (Eqn. 2) for a 4-layer ReLU MLP with one scalar input. This kernel is not shift-invariant and does not have a strong diagonal, making it poorly suited for kernel regression in low-dimensional problems. (b) A basic input mapping $\gamma(v) = [\cos 2\pi v, \sin 2\pi v]^{\mathrm{T}}$ makes the composed NTK $k_{\mathrm{NTK}}(\gamma(v_i), \gamma(v_j))$ shift-invariant (stationary). (c) A Fourier feature input mapping (Eqn. 5) can be used to tune the composed kernel's width, where we set $a_j = 1/j^p$ and $b_j = j$ for $j = 1, \ldots, n/2$. (d) Higher frequency mappings (lower $p$) result in composed kernels with wider spectra, which enables faster convergence for high-frequency components (see Figure 3).

After computing the Fourier features for our input points, we pass them through an MLP to get $f(\gamma(\mathbf{v}); \theta)$. As discussed previously, the result of training a network can be approximated by kernel regression using the kernel $h_{\mathrm{NTK}}(\mathbf{x}_i^{\mathrm{T}} \mathbf{x}_j)$. In our case, $\mathbf{x}_i = \gamma(\mathbf{v}_i)$ so the composed kernel becomes:

$$h_{\mathrm{NTK}}(\mathbf{x}_i^{\mathrm{T}} \mathbf{x}_j) = h_{\mathrm{NTK}}\left(\gamma\left(\mathbf{v}_i\right)^{\mathrm{T}} \gamma\left(\mathbf{v}_j\right)\right) = h_{\mathrm{NTK}}\left(h_\gamma\left(\mathbf{v}_i - \mathbf{v}_j\right)\right). \tag{8}$$

Thus, training a network on these embedded input points corresponds to kernel regression with the *stationary* composed NTK function $h_{\mathrm{NTK}} \circ h_\gamma$. The MLP function approximates a convolution of the composed NTK with a weighted Dirac delta at each input training point $\mathbf{v}_i$:

$$\hat{f} = (h_{\mathrm{NTK}} \circ h_\gamma) * \sum_{i=1}^{n} w_i \delta_{\mathbf{v}_i} \tag{9}$$

where $\mathbf{w} = \mathbf{K}^{-1} \mathbf{y}$ (from Eqn. 1). This allows us to draw analogies to signal processing, where the composed NTK acts similarly to a reconstruction filter. In the next section, we show that the frequency decay of the composed NTK determines the behavior of the reconstructed signal.

## 5   Manipulating the Fourier Feature Mapping

Preprocessing the inputs to a coordinate-based MLP with a Fourier feature mapping creates a composed NTK that is not only stationary but also *tunable*. By manipulating the settings of the $a_j$ and $\mathbf{b}_j$ parameters in Eqn. 5, it is possible to dramatically change both the rate of convergence and the generalization behavior of the resulting network. In this section, we investigate the effects of the Fourier feature mapping in the setting of 1D function regression.

We train MLPs to learn signals $f$ defined on the interval $[0, 1)$. We sample $cn$ linearly spaced points on the interval, using every $c^{\mathrm{th}}$ point as the training set and the remaining points as the test set. Since our composed kernel function is stationary, evaluating it at linearly spaced points on a periodic domain makes the resulting kernel matrix circulant: it represents a convolution and is diagonalizable by the Fourier transform. Thus, we can compute the eigenvalues of the composed NTK matrix by simply taking the Fourier transform of a single row. All experiments are implemented in JAX [8] and the NTK functions are calculated automatically using the Neural Tangents library [33].

**Visualizing the composed NTK.** We first visualize how modifying the Fourier feature mapping changes the composed NTK. We set $b_j = j$ (full Fourier basis in 1D) and $a_j = 1/j^p$ for $j = 1, \ldots, n/2$. We use $p = \infty$ to denote the mapping $\gamma(v) = [\cos 2\pi v, \sin 2\pi v]^{\mathrm{T}}$ that simply wraps $[0, 1)$ around the unit circle (this is referred to as the "basic" mapping in later experiments). Figure 2 demonstrates the effect of varying $p$ on the composed NTK. By construction, lower $p$ values result in a slower falloff in the frequency domain and a correspondingly narrower kernel in the spatial domain.

**Effects of Fourier features on network convergence.** We generate ground truth 1D functions by sampling $cn$ values from a family with parameter $\alpha$ as follows: we sample a standard i.i.d. Gaussian

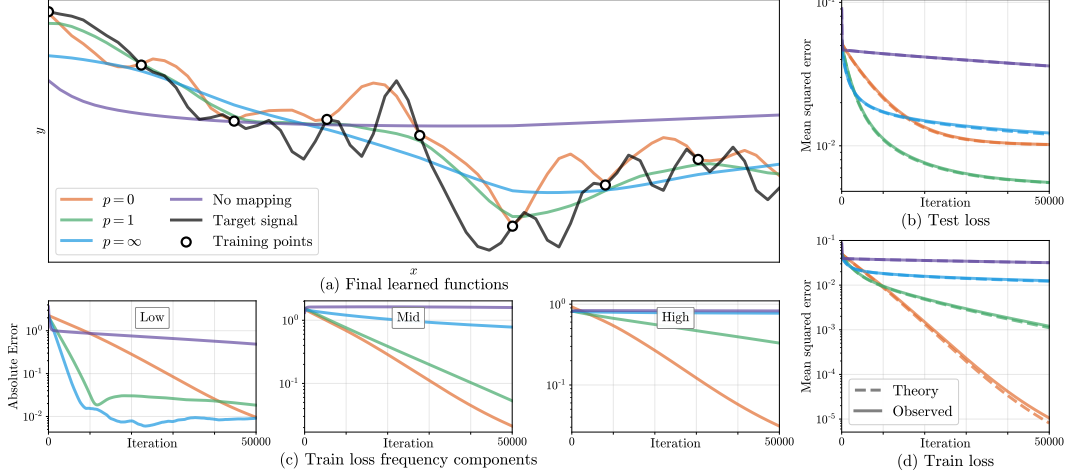

Figure 3: Combining a network with a Fourier feature mapping has dramatic effects on convergence and generalization. Here we train a network on 32 sampled points from a 1D function (a) using mappings shown in Fig. 2. A mapping with a smaller $p$ value yields a composed NTK with more power in higher frequencies, enabling the corresponding network to learn a higher frequency function. The theoretical and experimental training loss improves monotonically with higher frequency kernels (d), but the test-set loss is lowest at $p = 1$ and falls as the network starts to overfit (b). As predicted by Eqn. 4, we see roughly log-linear convergence of the training loss frequency components (c). Higher frequency kernels result in faster convergence for high frequency loss components, thereby overcoming the "spectral bias" observed when training networks with no input mapping.

vector of length $cn$, scale its $i^{th}$ entry by $1/i^\alpha$, then return the real component of its inverse Fourier transform. We will refer to this as a "$1/f^\alpha$ noise" signal.

In Figure 3, we train MLPs (4 layers, 1024 channels, ReLU activations) to fit a bandlimited $1/f^1$ noise signal ($c = 8, n = 32$) using Fourier feature mappings with different $p$ values. Figures 3b and 3d show that the NTK linear dynamics model accurately predict the effects of modifying the Fourier feature mapping parameters. Separating different frequency components of the training error in Figure 3c reveals that networks with narrower NTK spectra converge faster for low frequency components but essentially never converge for high frequency components, whereas networks with wider NTK spectra successfully converge across all components. The Fourier feature mapping $p = 1$ has adequate power across frequencies present in the target signal (so the network converges rapidly during training) but limited power in higher frequencies (preventing overfitting or aliasing).

**Tuning Fourier features in practice.** Eqn. 3 allows us to estimate a trained network's theoretical loss on a validation set using the composed kernel. For small 1D problems, we can minimize this loss with gradient-based optimization to choose mapping parameters $a_j$ (given a dense sampling of $b_j$). In this carefully controlled setting (1D signals, small training dataset, gradient descent with small learning rate, very wide networks), we find that this optimized mapping also achieves the best performance when training networks. Please refer to Appendix A.1 for details and experiments.

In real-world problems, especially in multiple dimensions, it is not feasible to use a feature mapping that densely samples Fourier basis functions; the number of Fourier basis functions scales with the number of training data points, which grows exponentially with dimension. Instead, we sample a set of random Fourier features [37] from a parametric distribution. We find that the exact sampling distribution family is much less important than the distribution's scale (standard deviation).

Figure 4 demonstrates this point using hyperparameter sweeps for a variety of sampling distributions. In each subfigure, we draw 1D target signals ($c = 2, n = 1024$) from a fixed $1/f^\alpha$ distribution and train networks to learn them. We use random Fourier feature mappings (of length 16) sampled from different distribution families (Gaussian, uniform, uniform in log space, and Laplacian) and sweep over each distribution's scale. Perhaps surprisingly, the standard deviation of the sampled frequencies alone is enough to predict test set performance, regardless of the underlying distribution's shape. We show that this holds for higher-dimensional tasks in Appendix A.4. We also observe that passing this

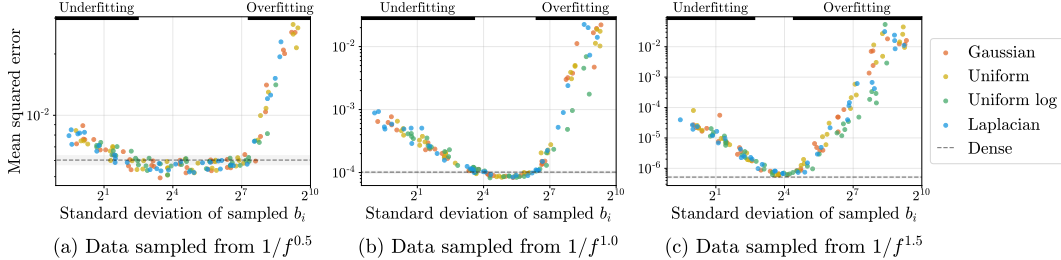

(a) Data sampled from $1/f^{0.5}$     (b) Data sampled from $1/f^{1.0}$     (c) Data sampled from $1/f^{1.5}$

Figure 4: We find that a sparse random sampling of Fourier features can perform as well as a dense set of features and that the width of the distribution matters more than the shape. Here, we generate random 1D signals from $1/f^\alpha$ noise and report the test-set accuracy of different trained models that use a sparse set (16 out of 1024) of random Fourier features sampled from different distributions. Each subplot represents a different family of 1D signals. Each dot represents a trained network, where the color indicates which Fourier feature sampling distribution is used. We plot the test error of each model versus the empirical standard deviation of its sampled frequencies. The best models using sparsely sampled features are able to match the performance of a model trained with dense Fourier features (dashed lines with error bars). All sampling distributions trace out the same curve, exhibiting underfitting (slow convergence) when the standard deviation of sampled frequencies is too low and overfitting when it is too high. This implies that the precise shape of the distribution used to sample frequencies does not have a significant impact on performance.

sparse sampling of Fourier features through an MLP matches the performance of using a dense set of Fourier features with the same MLP, suggesting a strategy for scaling to higher dimensions. We proceed with a Gaussian distribution for our higher-dimensional experiments in Section 6 and treat the scale as a hyperparameter to tune on a validation dataset.

# 6   Experiments

We validate the benefits of using Fourier feature mappings for coordinate-based MLPs with experiments on a variety of regression tasks relevant to the computer vision and graphics communities.

## 6.1   Compared mappings

In Table 1, we compare the performance of coordinate-based MLPs with no input mapping and with the following Fourier feature mappings ($\cos, \sin$ are applied elementwise):

**Basic:** $\gamma(\mathbf{v}) = [\cos(2\pi\mathbf{v}v), \sin(2\pi\mathbf{v})]^{\mathrm{T}}$. Simply wraps input coordinates around the circle.

**Positional encoding:** $\gamma(\mathbf{v}) = \left[\ldots, \cos(2\pi\sigma^{j/m}\mathbf{v}), \sin(2\pi\sigma^{j/m}\mathbf{v}), \ldots\right]^{\mathrm{T}}$ for $j = 0, \ldots, m-1$. Uses log-linear spaced frequencies for each dimension, where the scale $\sigma$ is chosen for each task and dataset by a hyperparameter sweep. This is a generalization of the "positional encoding" used by prior work [30, 43, 48]. Note that this mapping is deterministic and only contains on-axis frequencies, making it naturally biased towards data that has more frequency content along the axes.

**Gaussian:** $\gamma(\mathbf{v}) = [\cos(2\pi\mathbf{B}\mathbf{v}), \sin(2\pi\mathbf{B}\mathbf{v})]^{\mathrm{T}}$, where each entry in $\mathbf{B} \in \mathbb{R}^{m \times d}$ is sampled from $\mathcal{N}(0, \sigma^2)$, and $\sigma$ is chosen for each task and dataset with a hyperparameter sweep. In the absence of any strong prior on the frequency spectrum of the signal, we use an isotropic Gaussian distribution.

Our experiments show that all of the Fourier feature mappings improve the performance of coordinate-based MLPs over using no mapping and that the Gaussian RFF mapping performs best.

## 6.2   Tasks

We conduct experiments with direct regression, where supervision labels are in the same space as the network outputs, as well as indirect regression, where the network outputs are passed through a forward model to produce observations in the same space as the supervision labels (Appendix D contains a theoretical analysis of indirect regression through a linear forward model). For each task

| | Direct supervision | | | Indirect supervision | | | |
| | 2D image | | 3D shape [27] | 2D CT | | 3D MRI | 3D NeRF [30] |
| | Natural | Text | | Shepp | ATLAS | ATLAS | |
|---|---|---|---|---|---|---|---|
| No mapping | 19.32 | 18.40 | 0.864 | 16.75 | 15.44 | 26.14 | 22.41 |
| Basic | 21.71 | 20.48 | 0.892 | 23.31 | 16.95 | 28.58 | 23.16 |
| Positional enc. | 24.95 | 27.57 | 0.960 | 26.89 | 19.55 | 32.23 | 25.28 |
| Gaussian | **25.57** | **30.47** | **0.973** | **28.33** | **19.88** | **34.51** | **25.48** |

Table 1: We compare four different input mappings on a variety of low-dimensional regression tasks. All results are reported in PSNR except *3D shape*, which uses IoU (higher is better for all). *No mapping* represents using a standard MLP with no feature mapping. *Basic*, *Positional encoding*, and *Gaussian* are different variants of Fourier feature maps. For the *Direct supervision* tasks, the network is supervised using ground truth labels for each input coordinate. For the *Indirect supervision* tasks, the network outputs are passed through a forward model before the loss is applied (integral projection for CT, the Fourier transform for MRI, and nonlinear volume rendering for NeRF). Fourier feature mappings improve results across all tasks, with random Gaussian features performing best.

and dataset, we tune Fourier feature scales on a held-out set of signals. For each target signal, we train an MLP on a training subset of the signal and compute error over the remaining test subset. All tasks (except 3D shape regression) use L2 loss and a ReLU MLP with 4 layers and 256 channels. The 3D shape regression task uses cross-entropy loss and a ReLU MLP with 8 layers and 256 channels. We apply a sigmoid activation to the output for each task (except the view synthesis density prediction). We use 256 frequencies for the feature mapping in all experiments (see Appendix A.2 for experiments that investigate the effects of network depth and feature mapping sparsity). Appendix E provides additional details on each task and our implementations, and Appendix F shows more result figures.

**2D image regression.** In this task, we train an MLP to regress from a 2D input pixel coordinate to the corresponding RGB value of an image. For each test image, we train an MLP on a regularly-spaced grid containing $1/4$ of the pixels and report test error on the remaining pixels. We compare input mappings over a dataset of natural images and a dataset of text images.

**3D shape regression.** Occupancy Networks [27] implicitly represent a 3D shape as the "decision boundary" of an MLP, which is trained to output 0 for points outside the shape and 1 for points inside the shape. Each batch of training data is generated by sampling points uniformly at random from the bounding box of the shape and calculating their labels using the ground truth mesh. Test error is calculated using intersection-over-union versus ground truth on a set of points randomly sampled near the mesh surface to better highlight the different mappings' abilities to resolve fine details.

**2D computed tomography (CT).** In CT, we observe integral projections of a density field instead of direct measurements. In our 2D CT experiments, we train an MLP that takes in a 2D pixel coordinate and predicts the corresponding volume density at that location. The network is indirectly supervised by the loss between a sparse set of ground-truth integral projections and integral projections computed from the network's output. We conduct experiments using two datasets: procedurally-generated Shepp-Logan phantoms [40] and 2D brain images from the ATLAS dataset [23].

**3D magnetic resonance imaging (MRI).** In MRI, we observe Fourier transform coefficients of atomic response to radio waves under a magnetic field. In our 3D MRI experiments, we train an MLP that takes in a 3D voxel coordinate and predicts the corresponding response at that location. The network is indirectly supervised by the loss between a sparse set of ground-truth Fourier transform coefficients and Fourier transform coefficients computed from discretely querying the MLP on a voxel grid. We conduct experiments using the ATLAS dataset [23].

**3D inverse rendering for view synthesis.** In view synthesis, we observe 2D photographs of a 3D scene, reconstruct a representation of that scene, then render images from new viewpoints. To perform this task, we train a coordinate-based MLP that takes in a 3D location and outputs a color and volume density. This MLP is indirectly supervised by the loss between the set of 2D image observations and the same viewpoints re-rendered from the predicted scene representation. We use a simplified version of the method described in NeRF [30], where we remove hierarchical sampling and view dependence and replace the original positional encoding with our compared input mappings.

# 7 Conclusion

We leverage NTK theory to show that a Fourier feature mapping can make coordinate-based MLPs better suited for modeling functions in low dimensions, thereby overcoming the spectral bias inherent in coordinate-based MLPs. We experimentally show that tuning the Fourier feature parameters offers control over the frequency falloff of the combined NTK and significantly improves performance across a range of graphics and imaging tasks. These findings shed light on the burgeoning technique of using coordinate-based MLPs to represent 3D shapes in computer vision and graphics pipelines, and provide a simple strategy for practitioners to improve results in these domains.

## Broader Impact

This paper demonstrates how Fourier features can be used to enable coordinate-based MLPs to accurately model high-frequency functions in low-dimensional domains. Because we improve the performance of coordinate-based MLPs, we consider the impact of using those MLPs (such as those shown in Figure 1).

The 2D image regression case (Figure 1b) has historically been limited in its practical use to artistic image synthesis [15, 39] and synthesizing images designed to mislead classifiers [31]. It is difficult to quantify the potential societal impact of artistic image synthesis, but the ability to generate improved adversarial attacks on classifiers poses a risk in domains such as robotics or self-driving cars, and necessitates the continued study of how classifiers can be made robust to such attacks [26].

Given that coordinate-based MLPs have been shown to exhibit notable compression capabilities [30], advances in coordinate-based MLPs for image or video regression may also serve as a basis for an effective compression algorithm. Improved image compression may have positive value in terms of consumer photography and electronics experiences (expanded on-device or cloud storage), but may have potentially negative value by enabling easier private or governmental surveillance by making recordings easier to store for longer periods of time.

Improved performance of coordinate-based MLPs for CT and MRI imaging tasks (Figure 1d) may lead to improved medical imaging technologies, which generally have positive societal impacts: more accurate diagnoses and less expensive or more accessible diagnostic information for communities with limited access to medical services. However, given the serious impact an inaccurate medical diagnosis can have on a patient's well-being, the consequences of failure for this use case are significant.

Coordinate-based MLPs have also been used for 3D tasks such as predicting volume occupancy (Figure 1c) and view synthesis (Figure 1e) [27, 30]. Assessing the long-term impact of algorithms that reason about 3D occupancy is difficult, as this task is fundamental to much of perception and robotics and thus carries with it the broad potential upsides and downsides of increased automation [10]. But in the immediate future, improved view synthesis has salient positive effects: it may let filmmakers produce photorealistic effects and may allow for immersive 3D mapping applications or VR experiences. However, this progress may also inadvertently reduce employment opportunities for the human artists who currently produce these effects manually.

## Acknowledgements

We thank Ben Recht for advice, and Cecilia Zhang and Tim Brooks for their comments on the text. BM is funded by a Hertz Foundation Fellowship and acknowledges support from the Google BAIR Commons program. MT and SFK are funded by NSF Graduate Fellowships. RR was supported in part by ONR grants N000141712687 and N000142012529 and the Ronald L. Graham Chair. RN was supported in part by an FHL Vive Center Seed Grant. Google University Relations provided a generous donation of compute credits.

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
