[Supplementary Material 1]

# Supplementary Material For
# Fourier Features Let Networks Learn High Frequency Functions in Low Dimensional Domains

# Contents

## 1 Further experiments

### 1.1 Optimizing validation error through the NTK linear dynamics

Using Eqn. 3 in the main paper, we can predict what error a trained network will achieve on a set of testing points. Since this equation depends on the composed NTK, we can directly relate predicted test set loss to the Fourier feature mapping parameters $a$ and $b$ for a validation set of signals $\mathbf{y}_{val}$:

$$\mathcal{L}_{\text{opt}} = \left\| \mathbf{u}^{(t)} - \mathbf{y}_{\text{val}} \right\|_2^2 \approx \left\| \mathbf{K}_{\text{val}} \mathbf{K}^{-1} \left( \mathbf{I} - e^{-\eta \mathbf{K} t} \right) \mathbf{y} - \mathbf{y}_{\text{val}} \right\|_2^2, \tag{1}$$

where $\mathbf{K}_{\text{val}}$ is the composed NTK evaluated between points in a validation dataset $\mathbf{X}_{\text{val}}$ and training dataset $\mathbf{X}$, and $\eta$ and $t$ are the learning rate and number of iterations that will be used when training the actual network.

In Figure 1, we show the results of minimizing Eqn. 1 by gradient descent on $a_j$ values (with fixed corresponding "densely sampled" $b_j = j$) for validation sets sampled from three different $1/f^\alpha$ noise families. Note that gradient descent on this theoretical loss approximation produces $a_j$ values which are able to perform as well as the best "power law" $a_j$ values for each respective signal class (compared dashed lines versus $\times$ markers in Figure 1b). As mentioned in the main text, we find that this optimization strategy is only viable for small 1D regression problems. In our multidimensional tasks, using densely sampled $\mathbf{b}_j$ values is not tractable due to memory constraints. In addition, the theoretical approximation only holds when training the network using SGD, and in practice we train using the Adam optimizer [8].

(a) NTK Fourier spectrum

(b) Fourier features mapping performances

Figure 1: The Fourier feature mappings can be optimized for better performance on a class of target signals by using the linearized network approximation. Here we consider target signals sampled from three different power law distributions. In (a) we show the spectrum for composed kernels corresponding to different optimized feature mappings, where the feature mappings are initialized to match the "Power $\infty$" distribution. In (b) we take an alternative approach where we sweep over "power law" settings for our Fourier features. We find that tuning this simple parameterization is able to perform on par with the optimized feature maps.

### 1.2 Feature sparsity and network depth

In our experiments, we observe that deeper networks need fewer Fourier features than shallow networks. As the depth of the MLP increases, we observe that a sparser set of frequencies can achieve similar performance; Figure 2 illustrates this effect in the context of 2D image regression.

Again drawing on NTK theory, we understand this tradeoff as an effect of frequency "spreading," as illustrated in Figure 3. A Fourier featurization consists of only discrete frequencies, but when composed with the NTK, the influence of each discrete frequency "spreads" over its local neighborhood in the final spectrum. We find that the "spread" around each frequency feature increases for deeper networks. For an MLP to learn all of the frequency components in the target signal, its corresponding composed NTK must contain adequate power across the frequency support of the target signal. This is accomplished either by including more frequencies in the Fourier features or by spreading those frequencies through sufficient NTK depth.

Figure 2: In a 2D image regression task (explained in Section 5.1) we find that shallower networks require more Fourier features than deeper networks. This is explained by the frequency spreading effect shown in Figure 3. In this experiment we use the *Natural* image dataset and a Gaussian mapping. All of the network layers have 256 channels, and the networks are trained using an Adam [8] optimizer with a learning rate of $10^{-3}$.

(a) NTK Fourier spectrum with basic mapping

(b) NTK Fourier spectrum with basic mapping and an additional frequency

Figure 3: Each frequency included in a Fourier embedding is "spread" by the NTK, with deeper NTKs causing more frequency spreading. We posit that this frequency spreading is what enables an MLP with a sparse set of Fourier features to faithfully reconstruct a complex signal, which would be poorly reconstructed by either sparse Fourier feature regression or a plain coordinate-based MLP.

## 1.3 Gradient descent does not optimize Fourier features

One may wonder if the Fourier feature mapping parameters $a_j$ and $\mathbf{b}_j$ can be optimized alongside network weights using gradient descent, which may circumvent the need for careful initialization. We performed an experiment in which the $a_j, \mathbf{b}_j$ values are treated as trainable variables (along with the weights of the network) and optimize all variables with Adam to minimize training loss. Figure 4 shows that jointly optimizing these parameters does not improve performance compared to leaving them fixed.

(a) Train

(b) Test

Figure 4: "Training" the Fourier feature mapping parameters $a_j$ and $\mathbf{b}_j$ along with the network weights using Adam does not improve performance, as the $\mathbf{b}_j$ values do not deviate significantly from their initial values. We show that this holds when $\mathbf{b}_j$ are initialized at three different scales of Gaussian Fourier features in the case of the 2D image task ($a_j$ are always initialized as 1).

## 1.4   Visualizing underfitting and overfitting in 2D

Figure 4 in the main text shows (in a 1D setting) that as the scale of the Fourier feature sampling distribution increases, the trained network's error traces out a curve that starts in an underfitting regime (only low frequencies are learned) and ends in an overfitting regime (the learned function includes high-frequency detail not present in the training data). In Figure 5, we show analogous behavior for 2D image regression, demonstrating that the same phenomenon holds in a multidimensional problem. In Figure 6, we show how changing the scale for Gaussian Fourier features qualitatively affects the final result in the 2D image regression task.

(a) Test error for 2D image task        (b) Train and test error for 2D image task

Figure 5: An alternate version of Figure 4 from the main text where the underlying signal is a 2D image (see 2D image task details in Section 5.1) instead of 1D signal. This multi-dimensional case exhibits the same behavior as was seen in the 1D case: we see the same underfitting/overfitting pattern for four different isotropic Fourier feature distributions, and the distribution shape matters less than the scale of sampled $b_i$ values.

$\sigma = 1$      $\sigma = 2$      $\sigma = 10$      $\sigma = 32$      $\sigma = 64$

Figure 6: A visualization of the 2D image regression task with different Gaussian scales (corresponding to points along the curve shown in Figure 5). Low values of $\sigma$ underfit, resulting in oversmoothed interpolation, and large values of $\sigma$ overfit, resulting in noisy interpolation. We find that $\sigma = 10$ performs best for our *Natural* image dataset.

## 1.5   Failures of positional encoding (axis-aligned bias)

Here we present a simple experiment to directly showcase the benefits of using an isotropic frequency distribution, such as Gaussian RFF, compared to the axis-aligned "positional encoding" used in prior work [13, 16]. As discussed in the main paper, the positional encoding mapping only uses on-axis frequencies. This approach is well-suited to data that has more frequency content along the coordinate axes, but is not as effective for more natural signals.

In Figure 7, we conduct a simple 2D image experiment where we train a coordinate-based MLP (2 layers, 256 channels) to fit target 2D sinusoid images ($512 \times 512$ resolution). We sample 64 such 2D sinusoid images (regularly-sampled in polar coordinates, with 16 angles and 4 radii) and train a 2D coordinate-based MLP to fit each, using the same setup as the 2D image experiments described in Section 5.1. The isotropic Gaussian RFF mapping performs well across all angles, while the positional encoding mapping performs worse for frequencies that are not axis-aligned.

Figure 7: We train a coordinate-based MLP to fit target 2D images consisting of simple sinusoids at different frequencies and angles. The positional encoding mapping performs well at on-axis angles and performs worse on off-axis angles, while the Gaussian RFF mapping performs similarly well across all angles (results are averaged over radii). Error bars are plotted over runs with different randomly-sampled frequencies for the Gaussian RFF mapping, while positional encoding is deterministic.

# 2 Additional details for main text figures

## 2.1 Main text Figure 3 (effect of feature mapping on convergence speed)

In Figure 8, we present an alternate version of Figure 3 from the main text showing a denser sampling of $p$ values to better visualize the effect of changing Fourier feature falloff on the resulting trained network. Again, the feature mapping used here is $a_j = 1/j^p, b_j = j$ for $j = 1, \ldots, n/2$.

Figure 8: An extension of Figure 3 from the main paper, showing more values of $p$. In (c) we see that mappings with more gradual frequency falloff (lower $p$) converge significantly faster in mid and high frequencies, resulting in faster overall training convergence (d). In (b) we see that $p = 1$ achieves a lower test error than the other mappings.

## 2.2 Main text Figure 4 (different random feature distributions in 1D)

Exact details for the sampling distributions used to generate $b_j$ values for Figure 4 in the main text are shown in Table 1. In Figure 9, we present an alternate version showing both train and test performance, emphasizing the underfitting/overfitting regimes created by manipulating the scale of the Fourier features.

**Uniform log distribution** We include the *Uniform log* distribution because it is the random equivalent of the "positional encoding" sometimes used in prior work. One observation is that the sampling

for uniform-log variables ($X' = \sigma_{ul}^X$ where $X \sim \mathcal{U}[0,1)$) corresponds to the following CDF:

$$P(X' \leq x) = \frac{\log x}{\log \sigma_{ul}}, \quad \text{for } x \in [1, \sigma_{ul}), \tag{2}$$

which has the following PDF:

$$p(x) = \frac{d}{dx} P(X' \leq x) = \frac{1}{x \log \sigma_{ul}}. \tag{3}$$

This shows that the randomized equivalent of positional encoding is sampling from a distribution proportional to a $1/f$ falloff power law.

| Name | Sampled $b_j$ values |
|---|---|
| Gaussian | $\sigma_g X$ for $X \sim \mathcal{N}(0,1)$ |
| Uniform | $\sigma_u X$ for $X \sim \mathcal{U}[0,1)$ |
| Uniform log | $\sigma_{ul}^X$ for $X \sim \mathcal{U}[0,1)$ |
| Laplacian | $\sigma_l X$ for $X \sim \text{Laplace}(0,1)$ |
| Positional Enc. | $2^{\sigma_p X}$ for $X \in \text{linspace}(0,1)$ (deterministic) |

Table 1: Different distributions used for sampling frequencies, where $\sigma$ is each distribution's "scale".

(a) Data sampled from $\alpha = 0.5$     (b) Data sampled from $\alpha = 1.0$     (c) Data sampled from $\alpha = 1.5$

Figure 9: An alternate version of Figure 4 from the main text showing both training error and test error for a variety of different Fourier feature sampling distributions. Adding training error to the plot clearly distinguishes between the underfitting regime with low frequency $b_i$ (where train and test error are similar) versus the overfitting regime with high frequency $b_i$ (where the test error increases but training error approaches machine precision).

# 3 Stationary kernels

One of the primary benefits of our Fourier feature mapping is that it results in a *stationary* composed NTK function. In this section, we offer some intuition for why stationarity is desirable for our low-dimensional graphics and imaging problems.

First, let us consider the implications of using an MLP applied directly to a low-dimensional input (without any Fourier feature mapping). In this setting, the NTK is a function of the dot product between its inputs and of their norms [2, 3, 4, 7]. This makes the NTK *rotation*-invariant, but not *translation*-invariant. For our graphics and imaging applications, we want to be able to model an object or scene equally well regardless of its location, so translation-invariance or *stationarity* is a crucial property. We can then add approximate rotation invariance back by using an isotropic frequency sampling distribution.

This aligns with standard practice in signal processing, in which $k(\mathbf{u}, \mathbf{v}) = \tilde{h}(\mathbf{u} - \mathbf{v}) = \tilde{h}(\mathbf{v} - \mathbf{u})$ (*e.g.* the Gaussian or radial basis function kernel, or the sinc reconstruction filter kernel). This Euclidean notion of similarity based on difference vectors is better suited to the low-dimensional regime, in which we expect (and can afford) dense and nearly uniform sampling. Regression with a stationary kernel corresponds to reconstruction with a convolution filter: new predictions are sums of training points, weighted by a function of Euclidean distance.

One of the most important features of our sinusoidal input mapping is that it translates between these two regimes. If $\mathbf{u}, \mathbf{v} \in \mathbb{R}^d$ for small $d$, $\gamma$ is our Fourier feature embedding function, and $k$ is a dot

product kernel function, then $k(\gamma(\mathbf{u}), \gamma(\mathbf{v})) = h(\gamma(\mathbf{u})^{\mathrm{T}}\gamma(\mathbf{v})) = \tilde{h}(\mathbf{u} - \mathbf{v})$. In words, our sinusoidal input mapping transforms a dot product kernel into a stationary one, making it better suited to the low-dimensional regime.

This effect is illustrated in a simple 1D example in Figure 10, which shows that the benefits of a stationary composed NTK indeed appear in the MLP setting with a basic Fourier featurization (using a single frequency). We train MLPs with and without this basic Fourier embedding to learn a set of shifted 1D Gaussian probability density functions. The plain MLP successfully fits a zero-centered function but struggles to fit shifted functions, while the MLP with basic Fourier embedding exhibits stationary behavior, with good performance regardless of shifts.

(a) Example target signals  (b) Reconstruction accuracy

Figure 10: A plain coordinate-based MLP can learn a centered function (in this case a Gaussian density) but struggles to model shifts of the same function. Adding a basic Fourier embedding (with a single frequency) enables the MLP to fit the target function equally well regardless of shifts. The NTK corresponding to the plain MLP is based on dot products between inputs, whereas the NTK corresponding to the NTK with Fourier embedding is based on Euclidean distances between inputs, making it shift-invariant. In this experiment we train an MLP (4 layers, 256 channels, ReLU activation) for 500 iterations using the Adam [8] optimizer with a learning rate of $10^{-4}$. We report mean and standard deviation performance over 20 random network initializations.

## 4   Indirect supervision through a linear map

In some of the tasks we explore in this work, such as image regression or 3D shape regression, optimization is performed by minimizing a loss between the output of a network and a directly observed quantity, such as the color of a pixel or the occupancy of a voxel. But in many graphics and imaging applications of interest, measurements are *indirect*, and the loss must be computed on the output of a network after it has been processed by some physical forward model. In NeRF [13], measurements are taken by sampling and compositing along rays in each viewing direction. In MRI, measurements are taken along various curves through the frequency domain. In CT, measurements are integral projections of the subject at various angles, which correspond to measuring lines through the origin in the frequency domain. Although the measurement transformation for NeRF is nonlinear (in density, although it is linear in color), those for both CT and MRI are linear. In this section, we extend the linearized training dynamics of Lee *et al.* [9] to the setting of training through a linear operator denoted by a matrix $\mathbf{A}$. This allows us to modify Eqn. 3 to incorporate $\mathbf{A}$, thereby demonstrating that the conclusions drawn in this work for the "direct" regression case also apply to the "indirect" case.

Our derivation closely follows Lee *et al.* [9], and begins by replacing the neural network $f$ with its linearization around the initial parameters $\theta_0$:

$$f_t^{\mathrm{lin}}(\mathbf{x}) \triangleq f_0(\mathbf{x}) + \nabla_\theta f_0(\mathbf{x})|_{\theta=\theta_0}\omega_t \,, \tag{4}$$

where $\omega_t \triangleq \theta_t - \theta_0$ denotes the change in network parameters since initialization and $t$ denotes time in continuous-time gradient flow dynamics. Then [9] describes the dynamics of gradient flow:

$$\dot{f}_t^{\mathrm{lin}}(\mathbf{x}) = -\eta\hat{\Theta}_0(\mathbf{x}, \mathbf{X})\nabla_{f_t^{\mathrm{lin}}(\mathbf{X})}\mathcal{L} \,, \tag{5}$$

where $\hat{\Theta}_t(\cdot, \cdot) = \nabla_\theta f_t(\cdot)\nabla_\theta f_t(\cdot)^{\mathrm{T}}$ is the NTK matrix at time $t$ ($\hat{\Theta}_t$ is shorthand for $\hat{\Theta}_t(\mathbf{X}, \mathbf{X})$) and $\mathcal{L}$ is the training loss. At this point, we depart slightly from the analysis of [9]: instead of

$\mathcal{L} = \sum_{(\mathbf{x},y)\in\mathcal{D}} \ell(f_t^{\mathrm{lin}}(\mathbf{x}), y)$ we have $\mathcal{L} = \frac{1}{2}\left\|\mathbf{A}(f_t^{\mathrm{lin}}(\mathbf{X}) - \mathbf{y})\right\|_2^2$, where $\mathbf{y}$ denotes the vector of training labels. The gradient of the loss is then

$$\nabla_{f_t^{\mathrm{lin}}(\mathbf{x})}\mathcal{L} = \nabla_{f_t^{\mathrm{lin}}(\mathbf{x})}\frac{1}{2}\left\|\mathbf{A}\left(f_t^{\mathrm{lin}}(\mathbf{X}) - \mathbf{y}\right)\right\|_2^2 \tag{6}$$

$$= \mathbf{A}^{\mathrm{T}}\mathbf{A}\left(f_t^{\mathrm{lin}}(\mathbf{X}) - \mathbf{y}\right). \tag{7}$$

Substituting this into the gradient flow dynamics of Eqn. 5 gives us:

$$\dot{f}_t^{\mathrm{lin}}(\mathbf{x}) = -\eta\hat{\Theta}_0(\mathbf{x}, \mathbf{X})\mathbf{A}^{\mathrm{T}}\mathbf{A}\left(f_t^{\mathrm{lin}}(\mathbf{X}) - \mathbf{y}\right), \tag{8}$$

with corresponding solution:

$$f_t^{\mathrm{lin}}(\mathbf{X}) = \left(\mathbf{I} - e^{-\eta\hat{\Theta}_0\mathbf{A}^{\mathrm{T}}\mathbf{A}t}\right)\mathbf{y} + e^{-\eta\hat{\Theta}_0\mathbf{A}^{\mathrm{T}}\mathbf{A}t}f_0(\mathbf{X}). \tag{9}$$

Finally, again following [9], we can decompose $f_t^{\mathrm{lin}}(\mathbf{x}) = \mu_t(\mathbf{x}) + \gamma_t(\mathbf{x})$ at any test point $\mathbf{x}$, where

$$\mu_t(\mathbf{x}) = \hat{\Theta}_0(\mathbf{x}, \mathbf{X})\hat{\Theta}_0^{-1}\left(\mathbf{I} - e^{-\eta\hat{\Theta}_0\mathbf{A}^{\mathrm{T}}\mathbf{A}t}\right)\mathbf{y}, \tag{10}$$

$$\gamma_t(\mathbf{x}) = f_0(\mathbf{x}) - \hat{\Theta}_0(\mathbf{x}, \mathbf{X})\hat{\Theta}_0^{-1}\left(\mathbf{I} - e^{-\eta\hat{\Theta}_0\mathbf{A}^{\mathrm{T}}\mathbf{A}t}\right)f_0(\mathbf{X}). \tag{11}$$

Assuming our initialization is small, *i.e.*, $f_0(\mathbf{x}) \approx 0 \ \forall\mathbf{x}$, we can write our approximate linearized network output as:

$$f_t^{\mathrm{lin}}(\mathbf{x}) \approx \hat{\Theta}_0(\mathbf{x}, \mathbf{X})\hat{\Theta}_0^{-1}\left(\mathbf{I} - e^{-\eta\hat{\Theta}_0\mathbf{A}^{\mathrm{T}}\mathbf{A}t}\right)\mathbf{y}. \tag{12}$$

In our previous analysis, we work instead with the expected or infinite-width NTK matrix $\mathbf{K}$, which is fixed throughout training. Using this notation, we have

$$\hat{\mathbf{y}}^{(t)} \approx f_t^{\mathrm{lin}}(\mathbf{X}_{\mathrm{test}}) \approx \mathbf{K}_{\mathrm{test}}\mathbf{K}^{-1}\left(\mathbf{I} - e^{-\eta\mathbf{K}\mathbf{A}^{\mathrm{T}}\mathbf{A}t}\right)\mathbf{y}. \tag{13}$$

This is nearly identical to Eqn. 3 in the main paper, except that the convergence is governed by the spectrum of $\mathbf{K}\mathbf{A}^{\mathrm{T}}\mathbf{A}$ rather than $\mathbf{K}$ alone. If $\mathbf{A}$ is unitary, such as the Fourier transform matrix used in (densely sampled) MRI, then training should behave exactly as if we were training on direct measurements. However, if $\mathbf{A}$ is not full rank, then training will only affect the components with nonzero eigenvalues in $\mathbf{K}\mathbf{A}^{\mathrm{T}}\mathbf{A}$. In this more common scenario, we want to design a kernel that will provide large eigenvalues in the components that $\mathbf{A}$ can represent, so that the learnable components will converge quickly, and provide reasonable priors for the components we cannot learn.

In our two tasks that supervise through a linear map, CT and MRI, the $\mathbf{A}^{\mathrm{T}}\mathbf{A}$ has a structure that illuminates how the linear map interacts with the composed NTK. The $\mathbf{A}^{\mathrm{T}}\mathbf{A}$ matrices for both these tasks are diagonalizable by the DFT matrix, where the diagonal entries are simply the number of times the corresponding frequency is measured by the MRI or CT sampling patterns. This follows from the fact that CT and MRI measurements can both be formulated as Fourier space sampling: CT samples rotated slices in Fourier space through the origin [5] and MRI samples operator-chosen Fourier trajectories. This means that frequencies not observed by the MRI or CT sampling patterns will never be supervised during training. Therefore, it is crucial to choose a Fourier feature mapping that results in a composed NTK with a good prior on these frequencies.

## 5   Task details

We present additional details for each task from Section 6 in the main text, including training parameters, forward models, datasets, etc. All experiments are implemented using JAX [6] and trained on a single K80 or RTX2080Ti GPU. Training a single MLP took between 10 seconds (for the 2D image task) and 30 minutes (for the inverse rendering task).

### 5.1   2D image

The 2D image regression tasks presented in the main text all use $512 \times 512$ resolution images. A subsampled grid of $256 \times 256$ pixels is used as training data, and an offset grid of $256 \times 256$ pixels

is used for testing. We use two image datasets: *Natural* and *Text*, each consisting of 32 images. The *Natural* images are generated by taking center crops of randomly sampled images from the Div2K dataset [1]. The *Text* images are generated by placing random strings of text with random sizes and colors on a white background (examples can be seen in Figure 11). For each dataset we perform a hyperparameter sweep over feature mapping scales on 16 images. We find that scales $\sigma_g = 10$ and $\sigma_p = 6$ work best for the *Natural* dataset and $\sigma_g = 14$ and $\sigma_p = 5$ work best for the *Text* dataset (see Table 1 for mapping definitions). In Table 2, we report model performance using the optimal mapping scale on the remaining 16 images.

|  | Natural | Text |
|---|---|---|
| No mapping | $19.32 \pm 2.48$ | $18.40 \pm 2.23$ |
| Basic | $21.71 \pm 2.71$ | $20.48 \pm 1.96$ |
| Positional enc. | $24.95 \pm 3.72$ | $27.57 \pm 3.07$ |
| Gaussian | $\mathbf{25.57 \pm 4.19}$ | $\mathbf{30.47 \pm 2.11}$ |

Table 2: 2D image results (mean $\pm$ standard deviation of PSNR)

Each model (MLP with 4 layers, 256 channels, ReLU activation) is trained for 2000 iterations using the Adam [8] optimizer with default settings ($\beta_1 = 0.9$, $\beta_2 = 0.999$, $\epsilon = 10^{-8}$). Learning rates are manually tuned for each dataset and method. For *Natural* images a learning rate of $10^{-3}$ is used for the Gaussian RFF and the positional encoding, and a learning rate of $10^{-2}$ is used for the basic mapping and "no mapping" methods. For the *Text* images a learning rate of $10^{-3}$ is used for all methods.

## 5.2   3D shape

We evaluate the 3D shape regression task (similar to Occupancy Networks [11]) on four complex triangle meshes commonly used in computer graphics applications (*Dragon*, *Armadillo*, *Buddha*, and *Lucy*, shown in Figure 12), each containing hundreds of thousands of vertices. We train one coordinate-based MLP network to represent a single mesh rather than trying to generalize one network to encode multiple objects, since our goal is to demonstrate that a network with no mapping or the low frequency "basic" mapping cannot accurately represent even a *single* shape, let alone a whole class of objects.

We use a network with 8 layers of 256 channels each and a ReLU nonlinearity between each layer. Our batch size is $32^3$ points, and we use the Adam optimizer [8] with a learning rate starting at $5 \times 10^{-4}$ and exponentially decaying by a factor of $0.01$ over the course of 10000 total training iterations. At each training iteration, we sample a batch of 3D points uniformly at random from the bounding box of the mesh, and then calculate ground truth labels (using the point-in-mesh method implemented in the Trimesh library [12], which relies on the Embree kernel for acceleration [15]). We use cross-entropy loss to train the network to match these classification labels (0 for points outside the mesh, 1 for points inside).

The meshes are scaled to fit inside the unit cube $[0, 1]^3$ such that the centroid of the mesh is $(0.5, 0.5, 0.5)$. We use the *Lucy* statue mesh as a validation object to find optimal scale values for the positional encoding and Gaussian feature mapping. As described in the caption for Table 3, we calculate error on both a uniformly random test set and a test set that is close to the mesh surface (randomly chosen mesh vertices that have been perturbed by a random Gaussian vector with standard deviation 0.01) in order to illustrate that Fourier feature mappings provide a large benefit in resolving fine surface details. Both test sets have $64^3$ points.

In Figure 12, we visualize additional results on all four meshes mentioned above (including the validation mesh *Lucy*). We render normal maps, which are computed by taking the cross product of the numerical horizontal and vertical derivatives of the depth map. The original depth map is generated by intersecting camera rays with the first 0.5 isosurface of the network. We select the Fourier feature scales for (d) and (e) by doing a hyperparameter search based on validation loss for the *Lucy* mesh in the last row and report test loss over the other three meshes (Table 3). Note that the weights for each trained MLP are only 2MB, while the triangle mesh files for the objects shown are 61MB, 7MB, 79MB, and 32MB respectively.

|                 | Uniform points      | Boundary points     |
|-----------------|---------------------|---------------------|
| No mapping      | $0.959 \pm 0.006$   | $0.864 \pm 0.014$   |
| Basic           | $0.966 \pm 0.007$   | $0.892 \pm 0.017$   |
| Positional enc. | $0.987 \pm 0.005$   | $0.960 \pm 0.011$   |
| Gaussian        | $\mathbf{0.988 \pm 0.007}$ | $\mathbf{0.973 \pm 0.010}$ |

Table 3: 3D shape results (mean $\pm$ standard deviation of intersection-over-union). *Uniform points* is an "easy" test set where points are sampled uniformly at random from the bounding box of the ground truth mesh, while *Boundary points* is a "hard" test set where points are sampled near the boundary of the ground truth mesh.

## 5.3   2D CT

In computed tomography (CT), we observe measurements that are integral projections (integrals along parallel lines) of a density field. We construct a 2D CT task by using ground truth $512 \times 512$ resolution images, and computing 20 synthetic integral projections at evenly-spaced angles. For each of these images, the supervision data is the set of integral projections, and the test PSNR is evaluated over the original image.

We use two datasets for our 2D CT task: randomized Shepp-Logan phantoms [14], and the ATLAS brain dataset [10]. For each dataset, we perform a hyperparameter sweep over mapping scales on 8 examples. We found that scales $\sigma_g = 4$ and $\sigma_p = 3$ work best for the *Shepp* dataset and $\sigma_g = 5$ and $\sigma_p = 5$ work best for the *ATLAS* dataset. In Table 4, we report model performance using the optimal mapping scale on a distinct set of 8 images.

|                 | Shepp               | ATLAS               |
|-----------------|---------------------|---------------------|
| No mapping      | $16.75 \pm 3.64$    | $15.44 \pm 1.28$    |
| Basic           | $23.31 \pm 4.66$    | $16.95 \pm 0.72$    |
| Positional enc. | $26.89 \pm 1.46$    | $19.55 \pm 1.09$    |
| Gaussian        | $\mathbf{28.33 \pm 1.15}$ | $\mathbf{19.88 \pm 1.23}$ |

Table 4: 2D CT results (mean $\pm$ standard deviation of PSNR).

Each model (MLP with 4 layers, 256 channels, ReLU activation) is trained for 1000 iterations using the Adam [8] optimizer with default settings ($\beta_1 = 0.9$, $\beta_2 = 0.999$, $\epsilon = 10^{-8}$). The learning rate is manually tuned for each method. Gaussian RFF and positional encoding use a learning rate of $10^{-3}$, and the basic and "no mapping" method use a learning rate of $10^{-2}$.

## 5.4   3D MRI

In magnetic resonance imaging (MRI), we observe measurements that are Fourier coefficients of the atomic response to radio waves under a magnetic field. We construct a toy 3D MRI task by using ground truth $96 \times 96 \times 96$ resolution volumes and randomly sampling $\sim 13\%$ of the Fourier coefficients for each volume from an isotropic Gaussian. For each of these volumes, the supervision data is the set of sampled Fourier coefficients, and the test PSNR is evaluated over the original volume.

We use the ATLAS brain dataset [10] for our 3D MRI experiments. We perform a hyperparameter sweep over mapping scales on 6 examples. We find that scales $\sigma_g = 5$ and $\sigma_p = 4$ perform best. In Table 5, we report model performance using the optimal mapping scale on a distinct set of 6 images. Each model (MLP with 4 layers, 256 channels, ReLU activation) is trained for 1000 iterations using the Adam [8] optimizer with default settings ($\beta_1 = 0.9$, $\beta_2 = 0.999$, $\epsilon = 10^{-8}$). We use a manually-tuned learning rate of $2 \times 10^{-3}$ for each method. Results are visualized in Figure 14.

## 5.5   3D inverse rendering for view synthesis

In this task we use the "tiny NeRF" simplified version of the view synthesis method NeRF [13] where hierarchical sampling and view dependence have been removed. The model is trained to predict the color and volume density at an input 3D point. Volumetric rendering is used to render novel

|              | ATLAS            |
|--------------|------------------|
| No mapping   | $26.14 \pm 1.45$ |
| Basic        | $28.58 \pm 2.45$ |
| Positional enc. | $32.23 \pm 3.08$ |
| Gaussian     | $\mathbf{34.51 \pm 2.72}$ |

Table 5: 3D MRI results (mean $\pm$ standard deviation of PSNR).

viewpoints of the object. The loss is calculated between the rendered views and ground truth renders. In our experiments we use the NeRF *Lego* dataset of 120 images downsampled to $400 \times 400$ pixel resolution. The dataset is split into 100 training images, 7 validation images, and 13 test images. The reconstruction quality on the validation images is used to determine the best mapping scale; for this scene we find $\sigma_g = 6.05$ and $\sigma_p = 1.27$ perform best.

The model (MLP with 4 layers, 256 channels, ReLU activation) is trained for $5 \times 10^5$ iterations using the Adam [8] optimizer with default settings ($\beta_1 = 0.9$, $\beta_2 = 0.999$, $\epsilon = 10^{-8}$). The learning rate was manually tuned for each mapping: $10^{-2}$ for no mapping, $5 \times 10^{-3}$ for basic, $5 \times 10^{-4}$ for positional encoding, and $5 \times 10^{-4}$ for Gaussian. During training we use batches of 1024 rays.

The original NeRF method [13] uses an input mapping similar to the *Positional encoding* we compare against. The original NeRF mapping is smaller than our mappings (8 vs. 256 frequencies). We include metrics for this mapping in Table 6 under *Original pos. enc.* The positional encoding mappings only contain frequencies on the axes, and are therefore biased towards signals with on-axis frequency content (as demonstrated in Section 1.5). In our experiments we rotate the *Lego* scene, which was manually axis-aligned in the original dataset, for a more equitable comparison. Table 6 also reports metrics for positional encodings on the original axis-aligned scene. Results are visualized in Figure 15.

|                                  | 3D NeRF          |
|----------------------------------|------------------|
| No mapping                       | $22.41 \pm 0.92$ |
| Basic                            | $23.16 \pm 0.90$ |
| Original pos. enc.               | $24.81 \pm 0.88$ |
| Positional enc.                  | $25.28 \pm 0.83$ |
| Gaussian                         | $\mathbf{25.48 \pm 0.89}$ |
| Original pos. enc. (axis-aligned) | $25.60 \pm 0.76$ |
| Positional enc. (axis-aligned)   | $26.27 \pm 0.91$ |

Table 6: 3D NeRF results (mean and standard deviation of PSNR). Error is calculated based on held-out images of the scene since the ground truth radiance field is not known.

# 6 Additional results figures

(a) Ground Truth     (b) No mapping     (c) Basic     (d) Positional enc.     (e) Gaussian

Figure 11: Additional results for the 2D image regression task, for three images from our *Natural* dataset (top) and two images from our *Text* dataset (bottom).

(a) Ground Truth     (b) No mapping     (c) Basic     (d) Positional enc.     (e) Gaussian

Figure 12: Additional results for our 3D shape occupancy task.

(a) Ground Truth     (b) No mapping     (c) Basic     (d) Positional enc.     (e) Gaussian

Figure 13: Additional results for the 2D CT task.

(a) Ground Truth    (b) No mapping    (c) Basic    (d) Positional enc.    (e) Gaussian

Figure 14: Additional results for the 3D MRI task.

(a) Ground Truth    (b) No mapping    (c) Basic    (d) Positional enc.    (e) Gaussian

Figure 15: Additional results for the inverse rendering task [13].

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

[Supplementary Material 2]

## A Further experiments

### A.1 Optimizing validation error through the NTK linear dynamics

Using Eqn. 3 in the main paper, we can predict what error a trained network will achieve on a set of testing points. Since this equation depends on the composed NTK, we can directly relate predicted test set loss to the Fourier feature mapping parameters $a$ and $b$ for a validation set of signals $\mathbf{y}_{val}$:

$$\mathcal{L}_{\text{opt}} = \left\| \mathbf{u}^{(t)} - \mathbf{y}_{\text{val}} \right\|_2^2 \approx \left\| \mathbf{K}_{\text{val}} \mathbf{K}^{-1} \left( \mathbf{I} - e^{-\eta \mathbf{K} t} \right) \mathbf{y} - \mathbf{y}_{\text{val}} \right\|_2^2, \tag{10}$$

where $\mathbf{K}_{\text{val}}$ is the composed NTK evaluated between points in a validation dataset $\mathbf{X}_{\text{val}}$ and training dataset $\mathbf{X}$, and $\eta$ and $t$ are the learning rate and number of iterations that will be used when training the actual network.

In Figure 5, we show the results of minimizing Eqn. 10 by gradient descent on $a_j$ values (with fixed corresponding "densely sampled" $b_j = j$) for validation sets sampled from three different $1/f^\alpha$ noise families. Note that gradient descent on this theoretical loss approximation produces $a_j$ values which are able to perform as well as the best "power law" $a_j$ values for each respective signal class (compared dashed lines versus $\times$ markers in Figure 5b). As mentioned in the main text, we find that this optimization strategy is only viable for small 1D regression problems. In our multidimensional tasks, using densely sampled $\mathbf{b}_j$ values is not tractable due to memory constraints. In addition, the theoretical approximation only holds when training the network using SGD, and in practice we train using the Adam optimizer [21].

(a) NTK Fourier spectrum　　　(b) Fourier features mapping performances

Figure 5: The Fourier feature mappings can be optimized for better performance on a class of target signals by using the linearized network approximation. Here we consider target signals sampled from three different power law distributions. In (a) we show the spectrum for composed kernels corresponding to different optimized feature mappings, where the feature mappings are initialized to match the "Power $\infty$" distribution. In (b) we take an alternative approach where we sweep over "power law" settings for our Fourier features. We find that tuning this simple parameterization is able to perform on par with the optimized feature maps.

### A.2 Feature sparsity and network depth

In our experiments, we observe that deeper networks need fewer Fourier features than shallow networks. As the depth of the MLP increases, we observe that a sparser set of frequencies can achieve similar performance; Figure 6 illustrates this effect in the context of 2D image regression.

Again drawing on NTK theory, we understand this tradeoff as an effect of frequency "spreading," as illustrated in Figure 7. A Fourier featurization consists of only discrete frequencies, but when composed with the NTK, the influence of each discrete frequency "spreads" over its local neighborhood in the final spectrum. We find that the "spread" around each frequency feature increases for deeper networks. For an MLP to learn all of the frequency components in the target signal, its corresponding composed NTK must contain adequate power across the frequency support of the target signal. This is accomplished either by including more frequencies in the Fourier features or by spreading those frequencies through sufficient NTK depth.

Figure 6: In a 2D image regression task (explained in Section E.1) we find that shallower networks require more Fourier features than deeper networks. This is explained by the frequency spreading effect shown in Figure 7. In this experiment we use the *Natural* image dataset and a Gaussian mapping. All of the network layers have 256 channels, and the networks are trained using an Adam [21] optimizer with a learning rate of $10^{-3}$.

(a) NTK Fourier spectrum with basic mapping

(b) NTK Fourier spectrum with basic mapping and an additional frequency

Figure 7: Each frequency included in a Fourier embedding is "spread" by the NTK, with deeper NTKs causing more frequency spreading. We posit that this frequency spreading is what enables an MLP with a sparse set of Fourier features to faithfully reconstruct a complex signal, which would be poorly reconstructed by either sparse Fourier feature regression or a plain coordinate-based MLP.

## A.3 Gradient descent does not optimize Fourier features

One may wonder if the Fourier feature mapping parameters $a_j$ and $\mathbf{b}_j$ can be optimized alongside network weights using gradient descent, which may circumvent the need for careful initialization. We performed an experiment in which the $a_j, \mathbf{b}_j$ values are treated as trainable variables (along with the weights of the network) and optimize all variables with Adam to minimize training loss. Figure 8 shows that jointly optimizing these parameters does not improve performance compared to leaving them fixed.

(a) Train

(b) Test

Figure 8: "Training" the Fourier feature mapping parameters $a_j$ and $\mathbf{b}_j$ along with the network weights using Adam does not improve performance, as the $\mathbf{b}_j$ values do not deviate significantly from their initial values. We show that this holds when $\mathbf{b}_j$ are initialized at three different scales of Gaussian Fourier features in the case of the 2D image task ($a_j$ are always initialized as 1).

## A.4 Visualizing underfitting and overfitting in 2D

Figure 4 in the main text shows (in a 1D setting) that as the scale of the Fourier feature sampling distribution increases, the trained network's error traces out a curve that starts in an underfitting regime (only low frequencies are learned) and ends in an overfitting regime (the learned function includes high-frequency detail not present in the training data). In Figure 9, we show analogous behavior for 2D image regression, demonstrating that the same phenomenon holds in a multidimensional problem. In Figure 10, we show how changing the scale for Gaussian Fourier features qualitatively affects the final result in the 2D image regression task.

(a) Test error for 2D image task

(b) Train and test error for 2D image task

Figure 9: An alternate version of Figure 4 from the main text where the underlying signal is a 2D image (see 2D image task details in Section E.1) instead of 1D signal. This multi-dimensional case exhibits the same behavior as was seen in the 1D case: we see the same underfitting/overfitting pattern for four different isotropic Fourier feature distributions, and the distribution shape matters less than the scale of sampled $b_i$ values.

$\sigma = 1$     $\sigma = 2$     $\sigma = 10$     $\sigma = 32$     $\sigma = 64$

Figure 10: A visualization of the 2D image regression task with different Gaussian scales (corresponding to points along the curve shown in Figure 9). Low values of $\sigma$ underfit, resulting in oversmoothed interpolation, and large values of $\sigma$ overfit, resulting in noisy interpolation. We find that $\sigma = 10$ performs best for our *Natural* image dataset.

## A.5 Failures of positional encoding (axis-aligned bias)

Here we present a simple experiment to directly showcase the benefits of using an isotropic frequency distribution, such as Gaussian RFF, compared to the axis-aligned "positional encoding" used in prior work [30, 48]. As discussed in the main paper, the positional encoding mapping only uses on-axis frequencies. This approach is well-suited to data that has more frequency content along the coordinate axes, but is not as effective for more natural signals.

In Figure 11, we conduct a simple 2D image experiment where we train a coordinate-based MLP (2 layers, 256 channels) to fit target 2D sinusoid images ($512 \times 512$ resolution). We sample 64 such 2D sinusoid images (regularly-sampled in polar coordinates, with 16 angles and 4 radii) and train a 2D coordinate-based MLP to fit each, using the same setup as the 2D image experiments described in Section E.1. The isotropic Gaussian RFF mapping performs well across all angles, while the positional encoding mapping performs worse for frequencies that are not axis-aligned.

Figure 11: We train a coordinate-based MLP to fit target 2D images consisting of simple sinusoids at different frequencies and angles. The positional encoding mapping performs well at on-axis angles and performs worse on off-axis angles, while the Gaussian RFF mapping performs similarly well across all angles (results are averaged over radii). Error bars are plotted over runs with different randomly-sampled frequencies for the Gaussian RFF mapping, while positional encoding is deterministic.

# B  Additional details for main text figures

## B.1  Main text Figure 3 (effect of feature mapping on convergence speed)

In Figure 12, we present an alternate version of Figure 3 from the main text showing a denser sampling of $p$ values to better visualize the effect of changing Fourier feature falloff on the resulting trained network. Again, the feature mapping used here is $a_j = 1/j^p, b_j = j$ for $j = 1, \ldots, n/2$.

Figure 12: An extension of Figure 3 from the main paper, showing more values of $p$. In (c) we see that mappings with more gradual frequency falloff (lower $p$) converge significantly faster in mid and high frequencies, resulting in faster overall training convergence (d). In (b) we see that $p = 1$ achieves a lower test error than the other mappings.

## B.2  Main text Figure 4 (different random feature distributions in 1D)

Exact details for the sampling distributions used to generate $b_j$ values for Figure 4 in the main text are shown in Table 2. In Figure 13, we present an alternate version showing both train and test performance, emphasizing the underfitting/overfitting regimes created by manipulating the scale of the Fourier features.

**Uniform log distribution**    We include the *Uniform log* distribution because it is the random equivalent of the "positional encoding" sometimes used in prior work. One observation is that the sampling

for uniform-log variables ($X' = \sigma_{ul}^X$ where $X \sim \mathcal{U}[0,1)$) corresponds to the following CDF:

$$P(X' \leq x) = \frac{\log x}{\log \sigma_{ul}}, \quad \text{for } x \in [1, \sigma_{ul}), \tag{11}$$

which has the following PDF:

$$p(x) = \frac{d}{dx} P(X' \leq x) = \frac{1}{x \log \sigma_{ul}} . \tag{12}$$

This shows that the randomized equivalent of positional encoding is sampling from a distribution proportional to a $1/f$ falloff power law.

| Name | Sampled $b_j$ values |
|---|---|
| Gaussian | $\sigma_g X$ for $X \sim \mathcal{N}(0,1)$ |
| Uniform | $\sigma_u X$ for $X \sim \mathcal{U}[0,1)$ |
| Uniform log | $\sigma_{ul}^X$ for $X \sim \mathcal{U}[0,1)$ |
| Laplacian | $\sigma_l X$ for $X \sim \text{Laplace}(0,1)$ |
| Positional Enc. | $2^{\sigma_p X}$ for $X \in \text{linspace}(0,1)$ (deterministic) |

Table 2: Different distributions used for sampling frequencies, where $\sigma$ is each distribution's "scale".

(a) Data sampled from $\alpha = 0.5$   (b) Data sampled from $\alpha = 1.0$   (c) Data sampled from $\alpha = 1.5$

Figure 13: An alternate version of Figure 4 from the main text showing both training error and test error for a variety of different Fourier feature sampling distributions. Adding training error to the plot clearly distinguishes between the underfitting regime with low frequency $b_i$ (where train and test error are similar) versus the overfitting regime with high frequency $b_i$ (where the test error increases but training error approaches machine precision).

## C   Stationary kernels

One of the primary benefits of our Fourier feature mapping is that it results in a *stationary* composed NTK function. In this section, we offer some intuition for why stationarity is desirable for our low-dimensional graphics and imaging problems.

First, let us consider the implications of using an MLP applied directly to a low-dimensional input (without any Fourier feature mapping). In this setting, the NTK is a function of the dot product between its inputs and of their norms [3, 5, 6, 18]. This makes the NTK *rotation*-invariant, but not *translation*-invariant. For our graphics and imaging applications, we want to be able to model an object or scene equally well regardless of its location, so translation-invariance or *stationarity* is a crucial property. We can then add approximate rotation invariance back by using an isotropic frequency sampling distribution.

This aligns with standard practice in signal processing, in which $k(\mathbf{u}, \mathbf{v}) = \tilde{h}(\mathbf{u} - \mathbf{v}) = \tilde{h}(\mathbf{v} - \mathbf{u})$ (*e.g.* the Gaussian or radial basis function kernel, or the sinc reconstruction filter kernel). This Euclidean notion of similarity based on difference vectors is better suited to the low-dimensional regime, in which we expect (and can afford) dense and nearly uniform sampling. Regression with a stationary kernel corresponds to reconstruction with a convolution filter: new predictions are sums of training points, weighted by a function of Euclidean distance.

One of the most important features of our sinusoidal input mapping is that it translates between these two regimes. If $\mathbf{u}, \mathbf{v} \in \mathbb{R}^d$ for small $d$, $\gamma$ is our Fourier feature embedding function, and $k$ is a dot

product kernel function, then $k(\gamma(\mathbf{u}), \gamma(\mathbf{v})) = h(\gamma(\mathbf{u})^{\mathrm{T}}\gamma(\mathbf{v})) = \tilde{h}(\mathbf{u} - \mathbf{v})$. In words, our sinusoidal input mapping transforms a dot product kernel into a stationary one, making it better suited to the low-dimensional regime.

This effect is illustrated in a simple 1D example in Figure 14, which shows that the benefits of a stationary composed NTK indeed appear in the MLP setting with a basic Fourier featurization (using a single frequency). We train MLPs with and without this basic Fourier embedding to learn a set of shifted 1D Gaussian probability density functions. The plain MLP successfully fits a zero-centered function but struggles to fit shifted functions, while the MLP with basic Fourier embedding exhibits stationary behavior, with good performance regardless of shifts.

(a) Example target signals    (b) Reconstruction accuracy

Figure 14: A plain coordinate-based MLP can learn a centered function (in this case a Gaussian density) but struggles to model shifts of the same function. Adding a basic Fourier embedding (with a single frequency) enables the MLP to fit the target function equally well regardless of shifts. The NTK corresponding to the plain MLP is based on dot products between inputs, whereas the NTK corresponding to the NTK with Fourier embedding is based on Euclidean distances between inputs, making it shift-invariant. In this experiment we train an MLP (4 layers, 256 channels, ReLU activation) for 500 iterations using the Adam [21] optimizer with a learning rate of $10^{-4}$. We report mean and standard deviation performance over 20 random network initializations.

# D    Indirect supervision through a linear map

In some of the tasks we explore in this work, such as image regression or 3D shape regression, optimization is performed by minimizing a loss between the output of a network and a directly observed quantity, such as the color of a pixel or the occupancy of a voxel. But in many graphics and imaging applications of interest, measurements are *indirect*, and the loss must be computed on the output of a network after it has been processed by some physical forward model. In NeRF [30], measurements are taken by sampling and compositing along rays in each viewing direction. In MRI, measurements are taken along various curves through the frequency domain. In CT, measurements are integral projections of the subject at various angles, which correspond to measuring lines through the origin in the frequency domain. Although the measurement transformation for NeRF is nonlinear (in density, although it is linear in color), those for both CT and MRI are linear. In this section, we extend the linearized training dynamics of Lee *et al.* [22] to the setting of training through a linear operator denoted by a matrix $\mathbf{A}$. This allows us to modify Eqn. 3 to incorporate $\mathbf{A}$, thereby demonstrating that the conclusions drawn in this work for the "direct" regression case also apply to the "indirect" case.

Our derivation closely follows Lee *et al.* [22], and begins by replacing the neural network $f$ with its linearization around the initial parameters $\theta_0$:

$$f_t^{\mathrm{lin}}(\mathbf{x}) \triangleq f_0(\mathbf{x}) + \nabla_\theta f_0(\mathbf{x})|_{\theta=\theta_0}\omega_t \,, \tag{13}$$

where $\omega_t \triangleq \theta_t - \theta_0$ denotes the change in network parameters since initialization and $t$ denotes time in continuous-time gradient flow dynamics. Then [22] describes the dynamics of gradient flow:

$$\dot{f}_t^{\mathrm{lin}}(\mathbf{x}) = -\eta\hat{\Theta}_0(\mathbf{x}, \mathbf{X})\nabla_{f_t^{\mathrm{lin}}(\mathbf{X})}\mathcal{L} \,, \tag{14}$$

where $\hat{\Theta}_t(\cdot, \cdot) = \nabla_\theta f_t(\cdot)\nabla_\theta f_t(\cdot)^{\mathrm{T}}$ is the NTK matrix at time $t$ ($\hat{\Theta}_t$ is shorthand for $\hat{\Theta}_t(\mathbf{X}, \mathbf{X})$) and $\mathcal{L}$ is the training loss. At this point, we depart slightly from the analysis of [22]: instead of

$\mathcal{L} = \sum_{(\mathbf{x},y)\in\mathcal{D}} \ell(f_t^{\text{lin}}(\mathbf{x}), y)$ we have $\mathcal{L} = \frac{1}{2} \left\| \mathbf{A}(f_t^{\text{lin}}(\mathbf{X}) - \mathbf{y}) \right\|_2^2$, where $\mathbf{y}$ denotes the vector of training labels. The gradient of the loss is then

$$\nabla_{f_t^{\text{lin}}(\mathbf{x})}\mathcal{L} = \nabla_{f_t^{\text{lin}}(\mathbf{x})} \frac{1}{2} \left\| \mathbf{A} \left( f_t^{\text{lin}}(\mathbf{X}) - \mathbf{y} \right) \right\|_2^2 \tag{15}$$

$$= \mathbf{A}^{\text{T}}\mathbf{A} \left( f_t^{\text{lin}}(\mathbf{X}) - \mathbf{y} \right) . \tag{16}$$

Substituting this into the gradient flow dynamics of Eqn. 14 gives us:

$$\dot{f}_t^{\text{lin}}(\mathbf{x}) = -\eta\hat{\Theta}_0(\mathbf{x}, \mathbf{X})\mathbf{A}^{\text{T}}\mathbf{A} \left( f_t^{\text{lin}}(\mathbf{X}) - \mathbf{y} \right) , \tag{17}$$

with corresponding solution:

$$f_t^{\text{lin}}(\mathbf{X}) = \left( \mathbf{I} - e^{-\eta\hat{\Theta}_0\mathbf{A}^{\text{T}}\mathbf{A}t} \right) \mathbf{y} + e^{-\eta\hat{\Theta}_0\mathbf{A}^{\text{T}}\mathbf{A}t} f_0(\mathbf{X}) . \tag{18}$$

Finally, again following [22], we can decompose $f_t^{\text{lin}}(\mathbf{x}) = \mu_t(\mathbf{x}) + \gamma_t(\mathbf{x})$ at any test point $\mathbf{x}$, where

$$\mu_t(\mathbf{x}) = \hat{\Theta}_0(\mathbf{x}, \mathbf{X})\hat{\Theta}_0^{-1} \left( \mathbf{I} - e^{-\eta\hat{\Theta}_0\mathbf{A}^{\text{T}}\mathbf{A}t} \right) \mathbf{y} , \tag{19}$$

$$\gamma_t(\mathbf{x}) = f_0(\mathbf{x}) - \hat{\Theta}_0(\mathbf{x}, \mathbf{X})\hat{\Theta}_0^{-1} \left( \mathbf{I} - e^{-\eta\hat{\Theta}_0\mathbf{A}^{\text{T}}\mathbf{A}t} \right) f_0(\mathbf{X}) . \tag{20}$$

Assuming our initialization is small, *i.e.*, $f_0(\mathbf{x}) \approx 0 \; \forall\mathbf{x}$, we can write our approximate linearized network output as:

$$f_t^{\text{lin}}(\mathbf{x}) \approx \hat{\Theta}_0(\mathbf{x}, \mathbf{X})\hat{\Theta}_0^{-1} \left( \mathbf{I} - e^{-\eta\hat{\Theta}_0\mathbf{A}^{\text{T}}\mathbf{A}t} \right) \mathbf{y} . \tag{21}$$

In our previous analysis, we work instead with the expected or infinite-width NTK matrix $\mathbf{K}$, which is fixed throughout training. Using this notation, we have

$$\hat{\mathbf{y}}^{(t)} \approx f_t^{\text{lin}}(\mathbf{X}_{\text{test}}) \approx \mathbf{K}_{\text{test}}\mathbf{K}^{-1} \left( \mathbf{I} - e^{-\eta\mathbf{K}\mathbf{A}^{\text{T}}\mathbf{A}t} \right) \mathbf{y} . \tag{22}$$

This is nearly identical to Eqn. 3 in the main paper, except that the convergence is governed by the spectrum of $\mathbf{K}\mathbf{A}^{\text{T}}\mathbf{A}$ rather than $\mathbf{K}$ alone. If $\mathbf{A}$ is unitary, such as the Fourier transform matrix used in (densely sampled) MRI, then training should behave exactly as if we were training on direct measurements. However, if $\mathbf{A}$ is not full rank, then training will only affect the components with nonzero eigenvalues in $\mathbf{K}\mathbf{A}^{\text{T}}\mathbf{A}$. In this more common scenario, we want to design a kernel that will provide large eigenvalues in the components that $\mathbf{A}$ can represent, so that the learnable components will converge quickly, and provide reasonable priors for the components we cannot learn.

In our two tasks that supervise through a linear map, CT and MRI, the $\mathbf{A}^{\text{T}}\mathbf{A}$ has a structure that illuminates how the linear map interacts with the composed NTK. The $\mathbf{A}^{\text{T}}\mathbf{A}$ matrices for both these tasks are diagonalizable by the DFT matrix, where the diagonal entries are simply the number of times the corresponding frequency is measured by the MRI or CT sampling patterns. This follows from the fact that CT and MRI measurements can both be formulated as Fourier space sampling: CT samples rotated slices in Fourier space through the origin [7] and MRI samples operator-chosen Fourier trajectories. This means that frequencies not observed by the MRI or CT sampling patterns will never be supervised during training. Therefore, it is crucial to choose a Fourier feature mapping that results in a composed NTK with a good prior on these frequencies.

## E   Task details

We present additional details for each task from Section 6 in the main text, including training parameters, forward models, datasets, etc. All experiments are implemented using JAX [8] and trained on a single K80 or RTX2080Ti GPU. Training a single MLP took between 10 seconds (for the 2D image task) and 30 minutes (for the inverse rendering task).

### E.1   2D image

The 2D image regression tasks presented in the main text all use $512 \times 512$ resolution images. A subsampled grid of $256 \times 256$ pixels is used as training data, and an offset grid of $256 \times 256$ pixels

is used for testing. We use two image datasets: *Natural* and *Text*, each consisting of 32 images. The *Natural* images are generated by taking center crops of randomly sampled images from the Div2K dataset [1]. The *Text* images are generated by placing random strings of text with random sizes and colors on a white background (examples can be seen in Figure 15). For each dataset we perform a hyperparameter sweep over feature mapping scales on 16 images. We find that scales $\sigma_g = 10$ and $\sigma_p = 6$ work best for the *Natural* dataset and $\sigma_g = 14$ and $\sigma_p = 5$ work best for the *Text* dataset (see Table 2 for mapping definitions). In Table 3, we report model performance using the optimal mapping scale on the remaining 16 images.

|  | Natural | Text |
|---|---|---|
| No mapping | $19.32 \pm 2.48$ | $18.40 \pm 2.23$ |
| Basic | $21.71 \pm 2.71$ | $20.48 \pm 1.96$ |
| Positional enc. | $24.95 \pm 3.72$ | $27.57 \pm 3.07$ |
| Gaussian | $\mathbf{25.57 \pm 4.19}$ | $\mathbf{30.47 \pm 2.11}$ |

Table 3: 2D image results (mean $\pm$ standard deviation of PSNR)

Each model (MLP with 4 layers, 256 channels, ReLU activation, sigmoid output) is trained for 2000 iterations using the Adam [21] optimizer with default settings ($\beta_1 = 0.9$, $\beta_2 = 0.999$, $\epsilon = 10^{-8}$). Learning rates are manually tuned for each dataset and method. For *Natural* images a learning rate of $10^{-3}$ is used for the Gaussian RFF and the positional encoding, and a learning rate of $10^{-2}$ is used for the basic mapping and "no mapping" methods. For the *Text* images a learning rate of $10^{-3}$ is used for all methods.

## E.2  3D shape

We evaluate the 3D shape regression task (similar to Occupancy Networks [27]) on four complex triangle meshes commonly used in computer graphics applications (*Dragon*, *Armadillo*, *Buddha*, and *Lucy*, shown in Figure 16), each containing hundreds of thousands of vertices. We train one coordinate-based MLP network to represent a single mesh rather than trying to generalize one network to encode multiple objects, since our goal is to demonstrate that a network with no mapping or the low frequency "basic" mapping cannot accurately represent even a *single* shape, let alone a whole class of objects.

We use a network with 8 layers of 256 channels each and a ReLU nonlinearity between each layer. We apply a sigmoid activation to the output. Our batch size is $32^3$ points, and we use the Adam optimizer [21] with a learning rate starting at $5 \times 10^{-4}$ and exponentially decaying by a factor of 0.01 over the course of 10000 total training iterations. At each training iteration, we sample a batch of 3D points uniformly at random from the bounding box of the mesh, and then calculate ground truth labels (using the point-in-mesh method implemented in the Trimesh library [28], which relies on the Embree kernel for acceleration [45]). We use cross-entropy loss to train the network to match these classification labels (0 for points outside the mesh, 1 for points inside).

The meshes are scaled to fit inside the unit cube $[0, 1]^3$ such that the centroid of the mesh is $(0.5, 0.5, 0.5)$. We use the *Lucy* statue mesh as a validation object to find optimal scale values for the positional encoding and Gaussian feature mapping. As described in the caption for Table 4, we calculate error on both a uniformly random test set and a test set that is close to the mesh surface (randomly chosen mesh vertices that have been perturbed by a random Gaussian vector with standard deviation 0.01) in order to illustrate that Fourier feature mappings provide a large benefit in resolving fine surface details. Both test sets have $64^3$ points.

In Figure 16, we visualize additional results on all four meshes mentioned above (including the validation mesh *Lucy*). We render normal maps, which are computed by taking the cross product of the numerical horizontal and vertical derivatives of the depth map. The original depth map is generated by intersecting camera rays with the first 0.5 isosurface of the network. We select the Fourier feature scales for (d) and (e) by doing a hyperparameter search based on validation loss for the *Lucy* mesh in the last row and report test loss over the other three meshes (Table 4). Note that the weights for each trained MLP are only 2MB, while the triangle mesh files for the objects shown are 61MB, 7MB, 79MB, and 32MB respectively.

|  | Uniform points | Boundary points |
|---|---|---|
| No mapping | $0.959 \pm 0.006$ | $0.864 \pm 0.014$ |
| Basic | $0.966 \pm 0.007$ | $0.892 \pm 0.017$ |
| Positional enc. | $0.987 \pm 0.005$ | $0.960 \pm 0.011$ |
| Gaussian | $\mathbf{0.988 \pm 0.007}$ | $\mathbf{0.973 \pm 0.010}$ |

Table 4: 3D shape results (mean $\pm$ standard deviation of intersection-over-union). *Uniform points* is an "easy" test set where points are sampled uniformly at random from the bounding box of the ground truth mesh, while *Boundary points* is a "hard" test set where points are sampled near the boundary of the ground truth mesh.

### E.3 2D CT

In computed tomography (CT), we observe measurements that are integral projections (integrals along parallel lines) of a density field. We construct a 2D CT task by using ground truth $512 \times 512$ resolution images, and computing 20 synthetic integral projections at evenly-spaced angles. For each of these images, the supervision data is the set of integral projections, and the test PSNR is evaluated over the original image.

We use two datasets for our 2D CT task: randomized Shepp-Logan phantoms [40], and the ATLAS brain dataset [23]. For each dataset, we perform a hyperparameter sweep over mapping scales on 8 examples. We found that scales $\sigma_g = 4$ and $\sigma_p = 3$ work best for the *Shepp* dataset and $\sigma_g = 5$ and $\sigma_p = 5$ work best for the *ATLAS* dataset. In Table 5, we report model performance using the optimal mapping scale on a distinct set of 8 images.

|  | Shepp | ATLAS |
|---|---|---|
| No mapping | $16.75 \pm 3.64$ | $15.44 \pm 1.28$ |
| Basic | $23.31 \pm 4.66$ | $16.95 \pm 0.72$ |
| Positional enc. | $26.89 \pm 1.46$ | $19.55 \pm 1.09$ |
| Gaussian | $\mathbf{28.33 \pm 1.15}$ | $\mathbf{19.88 \pm 1.23}$ |

Table 5: 2D CT results (mean $\pm$ standard deviation of PSNR).

Each model (MLP with 4 layers, 256 channels, ReLU activation, sigmoid output) is trained for 1000 iterations using the Adam [21] optimizer with default settings ($\beta_1 = 0.9$, $\beta_2 = 0.999$, $\epsilon = 10^{-8}$). The learning rate is manually tuned for each method. Gaussian RFF and positional encoding use a learning rate of $10^{-3}$, and the basic and "no mapping" method use a learning rate of $10^{-2}$.

### E.4 3D MRI

In magnetic resonance imaging (MRI), we observe measurements that are Fourier coefficients of the atomic response to radio waves under a magnetic field. We construct a toy 3D MRI task by using ground truth $96 \times 96 \times 96$ resolution volumes and randomly sampling $\sim 13\%$ of the Fourier coefficients for each volume from an isotropic Gaussian. For each of these volumes, the supervision data is the set of sampled Fourier coefficients, and the test PSNR is evaluated over the original volume.

We use the ATLAS brain dataset [23] for our 3D MRI experiments. We perform a hyperparameter sweep over mapping scales on 6 examples. We find that scales $\sigma_g = 5$ and $\sigma_p = 4$ perform best. In Table 6, we report model performance using the optimal mapping scale on a distinct set of 6 images. Each model (MLP with 4 layers, 256 channels, ReLU activation, sigmoid output) is trained for 1000 iterations using the Adam [21] optimizer with default settings ($\beta_1 = 0.9$, $\beta_2 = 0.999$, $\epsilon = 10^{-8}$). We use a manually-tuned learning rate of $2 \times 10^{-3}$ for each method. Results are visualized in Figure 18.

### E.5 3D inverse rendering for view synthesis

In this task we use the "tiny NeRF" simplified version of the view synthesis method NeRF [30] where hierarchical sampling and view dependence have been removed. The model is trained to predict the color and volume density at an input 3D point. Volumetric rendering is used to render novel

|  | ATLAS |
| --- | --- |
| No mapping | $26.14 \pm 1.45$ |
| Basic | $28.58 \pm 2.45$ |
| Positional enc. | $32.23 \pm 3.08$ |
| Gaussian | $\mathbf{34.51 \pm 2.72}$ |

Table 6: 3D MRI results (mean $\pm$ standard deviation of PSNR).

viewpoints of the object. The loss is calculated between the rendered views and ground truth renders. In our experiments we use the NeRF *Lego* dataset of 120 images downsampled to $400 \times 400$ pixel resolution. The dataset is split into 100 training images, 7 validation images, and 13 test images. The reconstruction quality on the validation images is used to determine the best mapping scale; for this scene we find $\sigma_g = 6.05$ and $\sigma_p = 1.27$ perform best.

The model (MLP with 4 layers, 256 channels, ReLU activation, sigmoid on RGB output) is trained for $5 \times 10^5$ iterations using the Adam [21] optimizer with default settings ($\beta_1 = 0.9$, $\beta_2 = 0.999$, $\epsilon = 10^{-8}$). The learning rate is manually tuned for each mapping: $10^{-2}$ for no mapping, $5 \times 10^{-3}$ for basic, $5 \times 10^{-4}$ for positional encoding, and $5 \times 10^{-4}$ for Gaussian. During training we use batches of 1024 rays.

The original NeRF method [30] uses an input mapping similar to the *Positional encoding* we compare against. The original NeRF mapping is smaller than our mappings (8 vs. 256 frequencies). We include metrics for this mapping in Table 7 under *Original pos. enc.*. The positional encoding mappings only contain frequencies on the axes, and are therefore biased towards signals with on-axis frequency content (as demonstrated in Section A.5). In our experiments we rotate the *Lego* scene, which was manually axis-aligned in the original dataset, for a more equitable comparison. Table 7 also reports metrics for positional encodings on the original axis-aligned scene. Results are visualized in Figure 19.

|  | 3D NeRF |
| --- | --- |
| No mapping | $22.41 \pm 0.92$ |
| Basic | $23.16 \pm 0.90$ |
| Original pos. enc. | $24.81 \pm 0.88$ |
| Positional enc. | $25.28 \pm 0.83$ |
| Gaussian | $\mathbf{25.48 \pm 0.89}$ |
| Original pos. enc. (axis-aligned) | $25.60 \pm 0.76$ |
| Positional enc. (axis-aligned) | $26.27 \pm 0.91$ |

Table 7: 3D NeRF results (mean and standard deviation of PSNR). Error is calculated based on held-out images of the scene since the ground truth radiance field is not known.

# F   Additional results figures

|         |             |         |                  |              |
|:-------:|:-----------:|:-------:|:----------------:|:------------:|
| (a) Ground Truth | (b) No mapping | (c) Basic | (d) Positional enc. | (e) Gaussian |

Figure 15: Additional results for the 2D image regression task, for three images from our *Natural* dataset (top) and two images from our *Text* dataset (bottom).

(a) Ground Truth    (b) No mapping    (c) Basic    (d) Positional enc.    (e) Gaussian

Figure 16: Additional results for the 3D shape occupancy task [27].

(a) Ground Truth    (b) No mapping    (c) Basic    (d) Positional enc.    (e) Gaussian

Figure 17: Results for the 2D CT task.

(a) Ground Truth     (b) No mapping     (c) Basic     (d) Positional enc.     (e) Gaussian

Figure 18: Additional results for the 3D MRI task.

(a) Ground Truth     (b) No mapping     (c) Basic     (d) Positional enc.     (e) Gaussian

Figure 19: Additional results for the inverse rendering task [30].



[Supplementary Material 3 · Code Readme.pdf]

# Setup

We provide the code as IPython notebooks. For ease of use, we recommend using Google Colab (colab.research.google.com) to run all of the notebooks except 3d_shape_occupancy.ipynb. The notebooks are designed to install libraries that the Colab environment is missing. To run in Colab, upload the file to colab.research.google.com and enable the GPU in the runtime settings.

Install the following libraries if you would prefer to use your own local environment:
- JAX (GPU)
- jaxlib
- neural-tangents
- tqdm
- Livelossplot
- imageio
- PIL
- cv2
- numpy
- matplotlib
- phantominator
- gdown

To run 3d_shape_occupancy.ipynb the additional libraries are necessary:
- Embree
- pyembree
- trimesh

# Included Files

1d_regression.ipynb: Main text Fig. 2,3 and supp. Fig. 8
1d_scatter_plots.ipynb : Main text Fig. 4 and supp. Fig. 9
1d_ntk_opt.ipynb : Supp. Fig. 1
2d_image_regression.ipynb: Main text Fig. 1, Table 1 and supp. Fig. 2,6,11 Table 2
2d_CT.ipynb: Main text Table 1 and supp. Fig. 13 Table 4
2d_MRI.ipynb: Main text Fig. 1, Table 1 and supp. Fig. 14 Table 5
3d_shape_occupancy.ipynb: Main text Fig. 1, Table 1, and supp. Fig. 12 Table 3
3d_simple_nerf: Main text Fig. 1, Table 1 and supp. Fig. 15 Table 6
Kernel_spreading.ipynb : Supp. Fig. 3
toy_stationary_ex.ipynb : Supp. Fig. 10
axis_aligned_ex.ipynb: Supp. Fig. 7