[Reviews · NeurIPS 2020]

Review 1

Summary and Contributions: Using tools from the neural tangent kernel (NTK) literature, the authors show that a standard multilayer perceptron fails to learn high frequencies both in theory and in practice. To overcome this spectral bias, they use a Fourier feature mapping to transform the effective NTK into a stationary kernel with a tunable bandwidth. The paper relies on applying the Fourier features work by Rahimi and Recht to approximate the NTK kernel. The main contributions of this paper are two fold: applying an existing seminal method to a new problem which leads to surprising and interesting findings of relevance to practitioners in deep learning; and 2) a detailed empirical study of the NTK (and its approximation) to several different image related applications .

Strengths: *** Post rebuttal update *** I have read the authors' response and am satisfied with way they addressed the points raised and the modifications they'll introduce based on my feedback. ********************************************** The paper contains very rich set of experiments; it is a very thorough study of the NTK kernel and its properties through an empirical analysis of image based applications. The contribution is novel and relevant to the emerging topic of NTK and its properties.

Weaknesses: The authors claim that a standard multilayer perception fails to learn high frequencies *in theory* -- the statement is made in the abstract and the rest of the paper. However, the claim on the theoretical properties is not well supported and justified. My understanding is that this is based on empirical studies and observations and there is no 'theory' per se that claims that multilayer perceptron can't learn high frequencies. Elaborating on this statement or rephrasing it to make it more limited in scope would be a welcome addition/modification to the paper.

Correctness: I did not find inaccurate claims, although there are some confusing details highlighted in the "Weaknesses" section above.

Clarity: Overall the paper is clearly written and is of very high quality in terms of clarity.

Relation to Prior Work: The paper cites relevant literature and draws connections and parallels as needed.

Reproducibility: Yes

Additional Feedback:


Review 2

Summary and Contributions: This paper considers the neural network representations of low dimension signals such as the pixels of 2D images and 3D shapes. Based on the Neural Tangent Kernel (NTK) theory of neural networks, this paper proposes to use Fourier Feature Mapping to preprocess low dimensional signals before they are fed into the MLP (Multilayer Perception) network, which makes the composite kernel not only stationary but also tunable. Experimental results demonstrate that, with same amount of training points (e,g. 1/4 of the pixels for 2D images), the reconstruction performance on 2D images and 3D shapes are significantly better using Fourier Feature Mapping than the standard MLP methods.

Strengths: Both the theoretical and practical contributions are strong.

Weaknesses: The proposed methods are limited to low dimensional signals. It will be interesting if the theory and methods can be generalised to high dimensional signals such as images (including all the pixels as one signal).

Correctness: The claims and methods are correct.

Clarity: Yes, this paper is well presented.

Relation to Prior Work: Yes, related works are discussed.

Reproducibility: Yes

Additional Feedback:


Review 3

Summary and Contributions: The author presents that a Fourier feature mapping allows the coordinate-based MLPs to be better suited for modeling functions in low dimensions. Specifically, the authors show that adjusting the Fourier feature parameters can control the frequency falloff of the combined NTK. The authors compare input mappings on a various low-dimensional regression tasks and achieved favorable performance.

Strengths: + The paper is well written. + The related work well categorizes and summarizes the previous work. + To provide theoretical analysis, the authors well explained the background of NTK theory in a concise manner. + The visualization of Figure2 provides an motivation and insight. + The authors provide a simple experiment showing that a basic input mapping makes the composed NTK shift-invariant. Moreover, they showed that a Fourier feature mapping can be used to tune the composed kernel's width (high-frequency mapping can enable faster convergence). + The paper provides strong empirical evidence on various low-dimensional regression tasks.

Weaknesses: Cannot find critical weakness

Correctness: The arguments and experiments are solid.

Clarity: The paper is easy to follow and well written.

Relation to Prior Work: The related work section well describes the distinct points from previous works.

Reproducibility: Yes

Additional Feedback: Based on the NTK theory, the authors provide a theoretical analysis on why the simple Fourier feature mapping can enhance the coordinate-based MLP significantly. The simple experiments and corresponding figures makes the reader easy to follow the main flow and concept. The experimental results shows the effectiveness of the proposals. --After rebuttal -- I will keep my initial rating.


Review 4

Summary and Contributions: This paper analyzes the theoretical properties of using Fourier Features to replace raw coordinates as input features to a neural tangent kernel (NTK). It also empirically proves the advantages on MLPs of doing so for toy functions, as well as a few image regression tasks. ====== Update after author responses ====== Appreciate the careful responses to my concerns. I think most of them are well addressed. Also I'd encourage the authors to incorporate such discussions into the final version to bring more clarity to the future audience. I'd raise my rating from 6 to 7.

Strengths: The theoretical analysis of Fourier features on NTK seems sound, although I haven't read the equations closely. Experiments clearly show the advantage of Fourier features.

Weaknesses: 1. How Fourier features accelerate NTK convergence in the high-frequency range? Did I overlook something or it's not analyzed? This is an essential theoretical support to the merits of Fourier features. 2. The theory part is limited to the behavior on NTK. I understand analyzing Fourier features on MLPs is highly difficult, but I'm a bit worried there would be a significant gap between NTK and the actual behavior of MLPs (although they are asymptotically equivalent). 3. Examples in Section 5 are limited to 1D functions, which are a bit toyish.

Correctness: The claims seem correct, and are well supported by experiments.

Clarity: In general this paper is well written. But sections 4 and 5 could be revised to make the logical flow clearer.

Relation to Prior Work: Prior work is properly mentioned and related.

Reproducibility: Yes

Additional Feedback: Suggest to put more visualizations, e.g. reconstructed images (could put in appendix). Failed cases are welcome.

[Author Response · NeurIPS 2020]

We thank the reviewers for their kind and thoughtful comments on our work. We especially appreciate the supportive feedback regarding our experimental contributions. Below, we respond to reviewer-specific comments.

**Reviewer 1**

Our claim that a standard multilayer perceptron "fails to learn high frequencies in theory" is based on the theoretical connection between the MLP's convergence rate and the eigenvalues of the NTK matrix. Standard MLPs have an NTK with such extreme decay in high frequency eigenvalues [3,4] that training them to learn high frequency function components is impractical, and adding an appropriate Fourier feature mapping speeds up convergence by orders of magnitude. It is true that with unbounded training time, a standard MLP could eventually fit to high frequency components; we will revise the text to clarify that we are referring to convergence speed, not representational power in the limit of training time. For example, in the abstract, modifying "a standard MLP fails to learn high frequencies both in theory and in practice" to "a standard MLP has impractically slow convergence to high frequency signal components."

**Reviewer 2**

We suspect that the benefits of Fourier features may not be as dramatic in high dimensions, since it is harder to find a high-dimensional problem setting with dense observations where stationary kernels would be desirable. However, we agree that generalizations of this approach to higher dimensional data could be an exciting avenue for future work.

**Reviewer 4**

*1. Analysis of accelerated convergence:* Fourier features can be used to increase the eigenvalues of the NTK corresponding to higher frequencies, which directly determine convergence speed during training (Eqn. 4, taken from [1,2,15]). We show empirical examples of increasing high-frequency NTK eigenvalues in Figure 2. However, we do not analytically derive the eigenvalues of the NTK as a function of the Fourier features, which would be quite challenging (prior work has only been able to analytically derive NTK eigenvalues for shallow networks).

*2. NTK approximation:* You are correct that the NTK analysis is a statement about the limiting behavior of MLPs, in the limits of infinite width and infinitesimal SGD learning rate. However, we find that the convergence behavior of our trained MLPs in practice closely matches the predictions provided by the NTK theory for 1D problems. Figures 3b and 3d show that the loss curves predicted by NTK theory match those produced when training the corresponding MLPs.

In higher dimensions, we did not characterize to what degree the precise predictions of NTK theory transfer to MLPs; our focus was instead on using intuition from our 1D experiments to achieve high performance on these "real" higher-dimensional problems. A precise study of the applicability of NTK theory to MLPs in higher dimensions is an exciting avenue for future research.

*3. 1D toy examples:* Section 5 indeed contains intentionally toy 1D experiments in order to clearly illustrate frequency convergence effects and show that sparse Fourier features are sufficient (which is necessary in higher dimensions). We directly extend the 1D experiment in Figure 4 to a two-dimensional setting in Section 1.4 of the supplement, and demonstrate the same underfitting/overfitting phenomenon.

*Clarity of Sections 4 and 5:* Thanks for the suggestion. We agree these sections are very important to the flow of the paper and will revise the text to clarify the key takeaways.

*More visualizations of results:* Thanks for the suggestion. In addition to the 2D and 3D visual results in Section 6 of the appendix, we will also publish our code so that anyone can run experiments with our method on their own data.

[Meta-Review · NeurIPS 2020]

Using NTK theory the authors show that a standard multilayer perceptron fails to learn high frequencies both in theory and in practice. The authors then use a Fourier feature mapping to transform to overcome this bias. The experimental results also demonstrate that, with the same amount of training points (e,g. 1/4 of the pixels for 2D images), the reconstruction performance on 2D images and 3D shapes are significantly better using Fourier Feature Mapping than the standard MLP methods. All reviewers thought the paper contains a very rich set of experiments and interesting numerical results. The reviewers raised various technical concerns in their reviews but thought that the authors’ response adequately addressed these concerns and multiple reviewers raised their score. I concur with this assessment and recommend acceptance.